# PROTES: Probabilistic Optimization with Tensor Sampling

**Anastasia Batsheva**[*]
Skolkovo Institute of Science and Technology
Moscow, Russia
a.batsheva@skoltech.ru

**Andrei Chertkov**[*]
Skolkovo Institute of Science and Technology
and AIRI, Moscow, Russia
a.chertkov@skoltech.ru

**Gleb Ryzhakov**[*]
Skolkovo Institute of Science and Technology
Moscow, Russia
g.ryzhakov@skoltech.ru

**Ivan Oseledets**
Skolkovo Institute of Science and Technology
and AIRI, Moscow, Russia
i.oseledets@skoltech.ru

## Abstract

We developed a new method PROTES for black-box optimization, which is based on the probabilistic sampling from a probability density function given in the low-parametric tensor train format. We tested it on complex multidimensional arrays and discretized multivariable functions taken, among others, from real-world applications, including unconstrained binary optimization and optimal control problems, for which the possible number of elements is up to $2^{1000}$. In numerical experiments, both on analytic model functions and on complex problems, PROTES outperforms popular discrete optimization methods (Particle Swarm Optimization, Covariance Matrix Adaptation, Differential Evolution, and others).

## 1 Introduction

The multidimensional optimization problem is one of the most common in machine learning. It includes the important case of gradient-free discrete optimization [30, 20, 21, 41]:

$$\boldsymbol{x}_{min} = \min_{\boldsymbol{x}} \mathsf{f}(\boldsymbol{x}), \quad s.t. \ \boldsymbol{x} = [n_1, n_2, \ldots, n_d], \quad n_i \in \{1, 2, \ldots, N_i\}, \tag{1}$$

where $d$ is the dimensionality of the problem, and $N_1, N_2, \ldots, N_d$ are the numbers of items for each dimension. Such settings arise when searching for the minimum or maximum element in an implicitly given multidimensional array (tensor[1]), including when considering the discretization of functions from a continuous argument. Multidimensional discrete optimization problems are still computationally difficult in the case of complex target functions or large dimensions, and efficient direct gradient-free optimization procedures are highly needed.

The development of methods for low-rank tensor approximations has made it possible to introduce fundamentally new approaches for the approximation, storage, and operation with multidimensional

---

[*]Equal contribution.

[1]A tensor is a multidimensional array with several dimensions $d$ $(d \geq 1)$. A two-dimensional tensor $(d = 2)$ is a matrix, one-dimensional $(d = 1)$ - a vector. We use lower case and upper case (where this does not lead to confusion) letters for scalars, bold letters $(\boldsymbol{a}, \boldsymbol{b}, \boldsymbol{c}, \ldots)$ for vectors, upper case letters $(A, B, C, \ldots)$ for matrices, and calligraphic upper case letters $(\mathcal{A}, \mathcal{B}, \mathcal{C}, \ldots)$ for tensors with $d > 2$. The $(n_1, n_2, \ldots, n_d)$-th entry of a $d$-dimensional tensor $\mathcal{Y} \in \mathbb{R}^{N_1 \times N_2 \times \ldots \times N_d}$ is denoted by $y = \mathcal{Y}[n_1, n_2, \ldots, n_d]$, where $n_i = 1, 2, \ldots, N_i$ $(i = 1, 2, \ldots, d)$, and $N_i$ is a size of the $i$-th mode. The mode-$i$ slice of such tensor is denoted by $\mathcal{Y}[n_1, \ldots, n_{i-1}, :, n_{i+1}, \ldots, n_d]$.

37th Conference on Neural Information Processing Systems (NeurIPS 2023).

tensors [13, 6, 7, 40]. One of the common methods for low-rank approximation is the tensor train (TT) decomposition [26], which allows bypassing the curse of dimensionality. Many useful algorithms (e. g., element-wise addition, multiplication, solution of linear systems, convolution, integration, etc.) have effective implementations for tensors given in the TT-format. The complexity of these algorithms turns out to be polynomial in the dimension $d$ and the mode size $N$. It makes the TT-decomposition extremely popular in a wide range of applications, including computational mathematics [29, 2] and machine learning [31, 19].

In the last few years, new discrete optimization algorithms based on the TT-format have been proposed: TTOpt [37], Optima-TT [5], and several modifications [34, 35, 24, 36]. However, the development of new, more accurate, and robust TT-based methods for multidimensional discrete optimization is possible. In this work, we extend recent approaches for working with probability distributions and sampling in the TT-format [10, 25] to the optimization area. That is, we specify a multidimensional discrete probability distribution in the TT-format, followed by efficient sampling from it and updating[2] its parameters to approximate the optimum in a better way. This makes it possible to build an effective optimization method, and the contributions of our work are as follows:

- We develop a new method PROTES for optimization (finding the minimum or maximum[3] value) of multidimensional data arrays and discretized multivariable functions based on a sampling method from a probability distribution in the TT-format;
- We apply[4] PROTES for various analytic model functions and for several multidimensional QUBO and optimal control problems to demonstrate its performance and compare it with popular discrete optimization algorithms (Particle Swarm Optimization, Covariance Matrix Adaptation, Differential Evolution, and NoisyBandit) as well as TT-based methods (TTOpt and Optima-TT). We used the same set of hyperparameters of our algorithm for all experiments and obtained the best result for 19 of the 20 problems considered.

## 2   Optimization with probabilistic sampling

Our problem is to minimize the given multivariable discrete black-box function $\mathsf{f}$ (1). It can be formulated in terms of the multi-index search in an implicitly given $d$-dimensional tensor

$$\mathcal{Y} \in \mathbb{R}^{N_1 \times N_2 \times \ldots \times N_d}, \quad \mathcal{Y}[n_1, n_2, \ldots, n_d] = \mathsf{f}(\boldsymbol{x}), \quad \boldsymbol{x} = [n_1, n_2, \ldots, n_d],$$

for all $n_i = 1, 2, \ldots, N_i$ $(i = 1, 2, \ldots, d)$. The essence of our idea is to use a probabilistic method to find the minimum $\boldsymbol{x}_{min}$. We propose establishing a discrete distribution $\mathsf{p}(\boldsymbol{x})$ from which the minimum could be sampled with high probability. This distribution can be specified as a tensor $\mathcal{P}_\theta \in \mathbb{R}^{N_1 \times N_2 \times \ldots \times N_d}$ in some low-parametric representation with a set of parameters $\theta$, having the same shape as the target tensor $\mathcal{Y}$.

We start from a random non-negative tensor $\mathcal{P}_\theta$ and iteratively perform the following steps until the budget is exhausted or until convergence (see graphic illustration in Figure 1):

1. **Sample** $K$ candidates of $\boldsymbol{x}_{min}$ from the current distribution $\mathcal{P}_\theta$: $\mathcal{X}_K = \{\boldsymbol{x}_1, \boldsymbol{x}_2, \ldots, \boldsymbol{x}_K\}$;
2. **Compute** the corresponding function values: $y_1 = \mathsf{f}(\boldsymbol{x}_1), y_2 = \mathsf{f}(\boldsymbol{x}_2), \ldots, y_K = \mathsf{f}(\boldsymbol{x}_K)$;
3. **Select** $k$ best candidates with indices $\mathcal{S} = \{s_1, s_2, \ldots, s_k\}$ from $\mathcal{X}_K$ with the minimal objective value, i. e., $y_j \leq y_J$ for all $j \in \mathcal{S}$ and $J \in \{1, 2, \ldots, K\} \setminus \mathcal{S}$;
4. **Update** the probability distribution $\mathcal{P}_\theta$ $(\theta \to \theta^{(new)})$ to increase the likelihood of selected candidates $\mathcal{S}$. We make several $(k_{gd})$ gradient ascent steps with the learning rate $\lambda$ for the tensor $\mathcal{P}_\theta$, using the following loss function

$$\widehat{L}_\theta(\{x_{s_1}, x_{s_2}, \ldots, x_{s_k}\}) = \sum_{i=1}^{k} \log\left(\mathcal{P}_\theta[\boldsymbol{x}_{s_i}]\right). \tag{2}$$

---

[2]As will be shown in the work, we use several gradient ascent iterations for the likelihood to update the parameters of the TT-decomposition. However, it is important to note that the proposed optimization method is gradient-free, that is, it does not use the internal structure and gradient of the objective function.

[3]Further, for concreteness, we will consider the minimization problem in this paper, while the proposed method can be applied to the discrete maximization problem without any significant modifications.

[4]The program code with numerical examples is available in the repository `https://github.com/anabatsh/PROTES`.

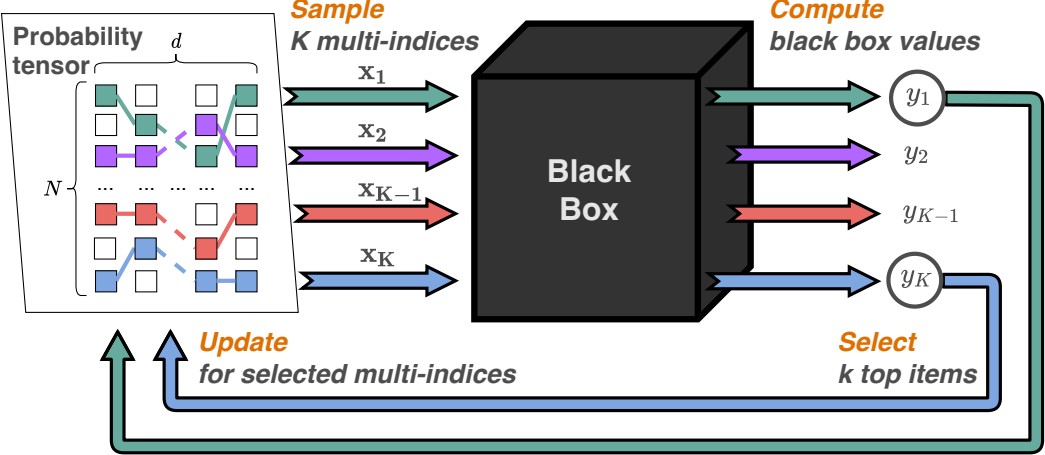

Figure 1: Schematic representation of the proposed optimization method PROTES.

After a sufficient number of iterations, we expect the tensor $\mathscr{P}_\theta$ to represent an almost Kronecker delta-function with a pronounced peak in the value of the minimum of the target function (or several peaks if the minimum is not unique). Therefore, this value will be sampled during the steps of our algorithm since the probability of sampling other values will be sufficiently small.

From a low-parameter representation $\mathscr{P}_\theta$ we expect an efficient sampling algorithm and efficient calculation procedure for the logarithms in (2) with automatic differentiation capability to enable gradient ascent methods. As will be shown below, the TT-representation of tensors satisfies these requirements. Further, we will omit the index $\theta$, assuming that the parameterized representation of the tensor $\mathscr{P}$ corresponds to the TT-format. Note that the values $K$, $k$, $k_{gd}$, $\lambda$ and the number of parameters in $\theta$ (i. e., rank of the TT-decomposition) are the hyperparameters of our algorithm.

## 3  Basic properties of the tensor train format

Let us dwell on the concept of the TT-format. A $d$-dimensional tensor $\mathscr{P} \in \mathbb{R}^{N_1 \times N_2 \times \ldots \times N_d}$ is said to be in the TT-format [26] if its elements are represented by the following formula

$$\mathscr{P}[n_1, n_2, \ldots, n_d] = \sum_{r_1=1}^{R_1} \sum_{r_2=1}^{R_2} \cdots \sum_{r_{d-1}=1}^{R_{d-1}} \mathscr{G}_1[1, n_1, r_1] \, \mathscr{G}_2[r_1, n_2, r_2] \, \ldots \, \mathscr{G}_d[r_{d-1}, n_d, 1], \quad (3)$$

where $(n_1, n_2, \ldots, n_d)$ is a multi-index ($n_i = 1, 2, \ldots, N_i$ for $i = 1, 2, \ldots, d$), integers $R_0, R_1, \ldots, R_d$ (with convention $R_0 = R_d = 1$) are named TT-ranks, and three-dimensional tensors $\mathscr{G}_i \in \mathbb{R}^{R_{i-1} \times N_i \times R_i}$ ($i = 1, 2, \ldots, d$) are named TT-cores. The TT-decomposition (3) (see also an illustration in Figure 2) allows to represent a tensor or a discretized multivariable function in a compact and descriptive low-parameter form, which is linear in dimension $d$, i. e., it has less than $d \cdot \max_{i=1,\ldots,d}(N_i R_i^2) \sim d \cdot \overline{N} \cdot \overline{R}^2$ parameters, where $\overline{N}$ and $\overline{R}$ are effective ("average") mode size and TT-rank respectively.

Linear algebra operations (e. g., element-wise addition, solution of linear systems, convolution, integration, etc.) on tensors in the TT-format, respectively, also have complexity linear in dimension if the TT-ranks are bounded. The TT-approximation for a tensor or discretized multivariable function may be built by efficient numerical methods, e. g., TT-SVD [27], TT-ALS [4], and TT-cross [28]. A detailed description of the TT-format and related algorithms are given in the works [26, 6, 7]. Below, we discuss only three operations in the TT-format, which will be used later in the work.

**Construction of the random TT-tensor.** In order to build a random non-negative TT-tensor of a given size $(N_1, N_2, \ldots, N_d)$ with a constant TT-rank $R$, it is enough to generate $d$ TT-cores $\mathscr{G}_1, \mathscr{G}_2, \ldots, \mathscr{G}_d$ (3-dimensional tensors) with random elements from the uniform distribution on the interval $(0, 1)$. We will refer to this method as tt_random($R, [N_1, N_2, \ldots, N_d]$).

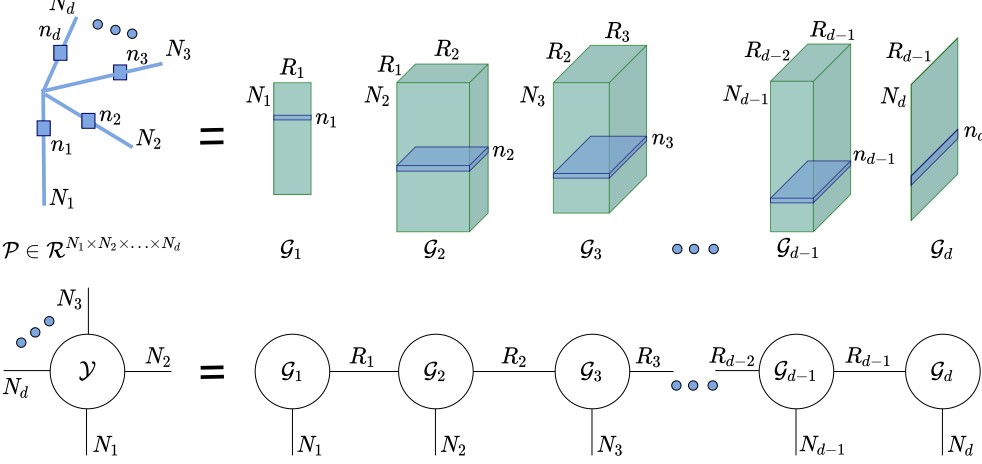

Figure 2: Schematic representation of the TT-decomposition. The top picture demonstrates the calculation of the specific tensor element $x = [n_1, n_2, \ldots, n_d]$ from its TT-representation, and the bottom picture presents the related tensor network diagram.

**Computation of the log-likelihood in the TT-format.** To calculate the logarithm $\log \mathscr{P}[x]$ in a given multi-index $x = (n_1, n_2, \ldots, n_d)$, we can use the basic formula (3) and then take the logarithm of the result. It can be shown that this operation has complexity $\mathcal{O}\left(d \cdot \overline{R}^2\right)$, because, roughly speaking, we $(d - 1)$ times multiply a vector of length $\overline{R}$ by a matrix of size $\overline{R} \times \overline{R}$ to get the result. The corresponding method will be called $\mathsf{tt\_log}(\mathscr{P}, x)$.

**Sampling from the tensor in the TT-format.** We use the approach proposed in the work [10] to generate a multi-index $x$ with a probability proportional to the corresponding value $p = \mathscr{P}[x]$ of the TT-tensor $\mathscr{P}$. The method is based on the sequential calculation of univariate conditional densities with efficient integration in the TT-format. The estimate for its complexity turns out to be the following: $\mathcal{O}\left(K \cdot d \cdot (\overline{N} + \overline{R}) \cdot \overline{R} + K \cdot d \cdot \alpha(\overline{N})\right)$, where $K$ is a number of requested samples, and $\alpha(n)$ is a complexity of sampling from generalized Bernoulli distribution with $n$ outcomes. Note that the algorithm allows sampling in the case of the initially non-normalized tensor, so we do not have to calculate the normalization factor. We will refer to this method as $\mathsf{tt\_sample}(\mathscr{P}, K)$.

## 4 Optimization method PROTES

With the formal scheme of the proposed approach given in Section 2 and the description of operations $\mathsf{tt\_random}$, $\mathsf{tt\_log}$, and $\mathsf{tt\_sample}$ given in Section 3, we can formulate our method PROTES for gradient-free discrete optimization in the TT-format, as presented in Algorithm 1. We denote as $\mathsf{adam}$, a function that performs $k_{gd}$ steps of gradient ascent for the TT-tensor $\mathscr{P}$ at multi-indices $\mathcal{X}$ by the well-known Adam method [18]. In this case, the learning rate is $\lambda$, the loss function is given in (2), and $\mathsf{tt\_log}$ with automatic differentiation support is used for the log-likelihood computation.

**Computational complexity of the method.** Let us estimate the complexity of the proposed algorithm, assuming that the number of requests to the target function (black-box) $M$ is fixed. With the known estimate for the complexity of the $\mathsf{tt\_sample}$ function, we can obtain the complexity of the sampling operations: $\mathcal{O}\left(\frac{M}{K} \cdot K \cdot d \cdot \left((\overline{N} + \overline{R}) \cdot \overline{R} + \alpha(\overline{N})\right)\right)$. Assuming that the complexity of one gradient step coincides with the complexity of calculating the differentiated function and using the estimate for the $\mathsf{tt\_sample}$ function, we can estimate the total complexity of the tensor updates: $\mathcal{O}\left(\frac{M}{K} \cdot k \cdot k_{gd} \cdot d \cdot \overline{R}^2\right)$. Combining the two above estimates we obtain the complexity of the method

$$\mathcal{O}\left(M \cdot d \cdot \left(\frac{k}{K} \cdot k_{gd} \cdot \overline{R}^2 + (\overline{N} + \overline{R}) \cdot \overline{R} + \alpha(N)\right)\right). \tag{4}$$

---

**Algorithm 1** Method PROTES in the TT-format for multidimensional discrete black-box optimization

---

**Data:** the function $f(\boldsymbol{x})$, that computes the value of the target tensor $\mathcal{Y} \in \mathbb{R}^{N_1 \times N_2 \times \ldots \times N_d}$ at the multi-index $\boldsymbol{x} = [n_1, n_2, \ldots, n_d]$; the maximum number of requests $M$; the number of generated samples per iteration $K$; the number of selected candidates per iteration $k$; the number of gradient ascent steps $k_{gd}$; the gradient ascent learning rate $\lambda$; the TT-rank of the probability tensor $R$.

**Result:** $d$-dimensional multi-index $\boldsymbol{x}_{min}$, which relates to the minimum value of the tensor $\mathcal{Y}$.

1   Initialize target multi-index and tensor value: $\boldsymbol{x}_{min} = \texttt{None}$, $y_{min} = \infty$
2   Generate random non-negative rank-$R$ TT-tensor: $\mathcal{P} = \textsf{tt\_random}\,(R, [N_1, N_2, \ldots, N_d])$
3   **for** $iter = 1$ **to** $M/K$ **do**
4      Generate $K$ samples from $\mathcal{P}$: $\boldsymbol{x}_1, \boldsymbol{x}_2, \ldots, \boldsymbol{x}_K = \textsf{tt\_sample}(\mathcal{P}, K)$
5      Compute related tensor values: $y_1 = f(\boldsymbol{x}_1), y_2 = f(\boldsymbol{x}_2), \ldots, y_K = f(\boldsymbol{x}_K)$
6      Find indices $\mathcal{S} = \{s_1, s_2, \ldots, s_k\}$ for top-$k$ minimum items in the list $[y_1, y_2, \ldots, y_K]$
7      Collect k-top candidates: $\mathcal{X} = \{\boldsymbol{x}_{s_1}, \boldsymbol{x}_{s_2}, \ldots, \boldsymbol{x}_{s_k}\}$, $\mathcal{Y} = \{y_{s_1}, y_{s_2}, \ldots, y_{s_k}\}$
8      If $\mathcal{Y}$ contains a value less than $y_{min}$, then update $\boldsymbol{x}_{min}$ and $y_{min}$
9      Update the TT-tensor: $\mathcal{P} \leftarrow \textsf{adam}(\mathcal{P}, \widehat{L}, \mathcal{X}, k_{gd}, \lambda)$ // *with the loss function* (2) *and the method* $\textsf{tt\_log}$
10   **return** $\underline{\boldsymbol{x}_{min}}$.

---

However, it is important to note that this estimate does not consider the complexity of calculating $M$ times the objective function $f$, which in practical applications can be significant and many times greater than the estimate (4).

**The intuition behind the method.**   The proposed method PROTES, like most gradient-free optimization approaches, is empirical; however, we can establish its connection with a well-known REINFORCE trick algorithm [43]. Let make a monotonic transformation $\mathsf{F}[f](\boldsymbol{x})$ of the target function $f$ to be minimized that transforms minimum to maximum. A reasonable choice for $\mathsf{F}[\cdot]$ is the Fermi-Dirac function

$$\mathsf{F}[f](\boldsymbol{x}) = \frac{1}{\exp\big((f(\boldsymbol{x}) - y_{\min} - E)/T\big) + 1},$$

where $y_{\min}$ is an exact or approximate minimum of $f$, $T > 0$ is a parameter and $E > 0$ is some small threshold. With the function $\mathsf{F}[f]$ we can find a maximum of the expectation $\max_\theta \mathsf{E}_{\xi_\theta}\mathsf{F}[f](\xi_\theta)$, where a family of random variables $\xi_\theta$ has a parameterised distribution function $\mathsf{p}_\theta(\boldsymbol{x})$. Using REINFORCE trick, we can estimate the gradient of the expectation by the following Monte-Carlo-like expression

$$\nabla_\theta \mathsf{E}_{\xi_\theta} \mathsf{F}[f](\xi_\theta) \approx \frac{1}{M} \sum_{i=1}^{M} \mathsf{F}[f](\boldsymbol{x}_i) \nabla_\theta \log \mathsf{p}_\theta(\boldsymbol{x}_i), \tag{5}$$

where $\{\boldsymbol{x}_i\}_1^M$ are independent realizations of the random variable $\xi_\theta$. If we find the optimal values of $\theta$, then we expect the optimal distribution $\mathsf{p}_\theta$ to peak at the maximum point for function $\mathsf{F}[f]$. Thus, we can obtain the argument of its maximum by sampling from this distribution. For very small values of $T$, only a few terms contribute to the sum (5), namely those $\boldsymbol{x}_i$ for which $f(\boldsymbol{x}_i) - y_{min} < E$ is hold. For these values of $\boldsymbol{x}$, $\mathsf{F}[f]$ is close to 1, while for the other samples its value is 0. Hence, we can discard all other samples and keep a few with the best values. So, we come to the loss function (2), where instead of the parameter $E$ we use a fixed number $k$ of the best samples, i. e., the samples for which the value of the target function $f$ is the smallest.

**Application of the method to constrained optimization.**   A very nice property of the proposed method is that it can be adapted to efficiently handle constraints such as a specified set of admissible multi-indices. One option is just to remove invalid samples from the top-k values, but in some cases, the probability of sampling multi-indices that are admissible is very low, so this approach will not work. Instead, if the constraint permits, we use the algorithm from the work [32] for the constructive building of tensors in the TT-format by a known analytic function, which defines the constraints. Once the indicator tensor (1 if the index is admissible and 0 if it is not) is built in the TT-format, we can just initialize the starting distribution $\mathcal{P}$ by it, and it will be guaranteed that the samples almost always belong to the admissible set.

# 5  Related work

Below, we briefly analyze classical approaches for discrete optimization and then discuss the methods based on the low-rank tensor approximations, which have become popular in recent years.

**Classical methods for gradient-free optimization.**   In many situations, the problem-specific target function is not differentiable, too complex, or its gradients are not helpful due to the non-convex nature of the problem [1], and standard well-known gradient-based methods cannot be applied directly. The examples include hyper-parameter selection, training neural networks with discrete weights, and policy optimization in reinforcement learning. In all these contexts, efficient direct gradient-free optimization procedures are highly needed. As for high-dimensional black-box optimization, evolutionary strategies (ES) [9] are one of the most advanced methods. This approach aims to optimize the parameters of the search distribution, typically a multidimensional Gaussian, to maximize the objective function. Finite difference schemes are commonly used to approximate gradients of the search distribution. Numerous works proposed techniques to improve the convergence of ES [23]; for example, second-order natural gradient [42] or the history of recent updates (Covariance Matrix Adaptation Evolution Strategy; CMA-ES) [15] may be used to generate updates. There is also a large variety of other heuristic methods for finding the global extremum. In particular, we note such popular approaches as NoisyBandit [33], Particle Swarm Optimization (PSO) [17], Simultaneous Perturbation Stochastic Approximation (SPSA) [22], Differential Evolution (DE) [38] and scrambled-Hammersley (scr-Hammersley) [14].

**Tensor-based methods for gradient-free optimization.**   Recently, the TT-decomposition has been actively used for multidimensional optimization. An iterative method TTOpt based on the maximum volume approach is proposed in the work [37]. TTOpt utilizes the theorem of sufficient proximity of the maximum modulo element of the submatrix having the maximum modulus of the determinant to the maximum modulo element of the tensor. Based on this observation, tensor elements are sampled from specially selected successive unfoldings of the tensor. Dynamic mapping of the tensor elements is carried out to find the minimum element, which converts the minimum values into maximum ones. The authors applied this approach to the problem of optimizing the weights of neural networks in the framework of reinforcement learning problems in [37] and to the QUBO problem in [24]. A similar optimization approach was also considered in [34] and [35]. One more promising algorithm, Optima-TT, was proposed in recent work [5]. This approach is based on the probabilistic sampling from the TT-tensor and makes it possible to obtain a very accurate approximation for the optimum of the given TT-tensor. However, this method is intended for directly optimizing the TT-tensors, which means that its success strongly depends on the quality of the TT-approximation for the original multidimensional data array. Therefore, one of the related methods in the TT-format (TT-SVD, TT-ALS, TT-cross, etc.) should be additionally used for approximation.

# 6  Numerical experiments

To evaluate the effectiveness of the proposed method, we carried out a series of 20 numerical experiments for various formulations of model problems. The results are presented in Table 1, where we report the approximation to the minimum value for each model problem (P-1, P-2, . . ., P-20) and all considered optimization methods (PROTES, BS-1, BS-2, . . ., BS-7). Taking into account the analysis of discrete optimization methods in the previous section as baselines we consider two tensor-based optimization methods: TTOpt[5] (BS1) and Optima-TT[6] (BS2), and five popular gradient-free optimization algorithms from the nevergrad framework [3]:[7] OnePlusOne (BS3), PSO (BS4), NoisyBandit (BS5), SPSA (BS6), and Portfolio approach (BS7), which is based on the combination of CMA-ES, DE, and scr-Hammersley methods. The model problems and obtained results will be discussed in detail below in this section.

For all the considered optimization problems, we used the default set of parameters for baselines, and for PROTES we fixed parameters as $K = 100$, $k = 10$, $k_{gd} = 1$, $\lambda = 0.05$, $R = 5$ (the description

---

[5]We used implementation of the method from `https://github.com/AndreiChertkov/ttopt`.

[6]We used implementation from `https://github.com/AndreiChertkov/teneva`. The TT-tensor for optimization was generated by the TT-cross method.

[7]See `https://github.com/facebookresearch/nevergrad`.

Table 1: Minimization results for all selected benchmarks (P-01 – P-20). The values obtained by the proposed method PROTES and by all considered baselines (BS1 – BS7) are reported.

| | | PROTES | BS-1 | BS-2 | BS-3 | BS-4 | BS-5 | BS-6 | BS-7 |
|---|---|---|---|---|---|---|---|---|---|
| ANALYTIC FUNCTIONS | P-01 | **1.3E+01** | **1.3E+01** | **1.3E+01** | **1.3E+01** | **1.3E+01** | 2.1E+01 | **1.3E+01** | **1.3E+01** |
| | P-02 | **6.5E+00** | **6.5E+00** | **6.5E+00** | 6.9E+00 | 6.8E+00 | 1.5E+01 | 7.5E+00 | 6.8E+00 |
| | P-03 | **-9.4E-01** | **-9.4E-01** | **-9.4E-01** | **-9.4E-01** | **-9.4E-01** | -3.5E-01 | **-9.4E-01** | **-9.4E-01** |
| | P-04 | **1.3E+00** | **1.3E+00** | **1.3E+00** | **1.3E+00** | **1.3E+00** | 6.3E+00 | **1.3E+00** | **1.3E+00** |
| | P-05 | **-3.7E+00** | **-3.7E+00** | **-3.7E+00** | -2.6E+00 | -3.0E+00 | -1.8E+00 | -1.2E+00 | **-3.7E+00** |
| | P-06 | **1.2E-01** | **1.2E-01** | **1.2E-01** | **1.2E-01** | **1.2E-01** | 1.3E-01 | 4.2E-01 | **1.2E-01** |
| | P-07 | **6.2E+06** | **6.2E+06** | **6.2E+06** | 6.3E+06 | 1.7E+07 | 2.2E+10 | 3.1E+08 | **6.2E+06** |
| | P-08 | **6.0E+01** | **6.0E+01** | **6.0E+01** | **6.0E+01** | **6.0E+01** | 1.2E+02 | 1.0E+02 | **6.0E+01** |
| | P-09 | **2.7E+00** | **2.7E+00** | **2.7E+00** | **2.7E+00** | **2.7E+00** | 2.9E+00 | 3.4E+00 | **2.7E+00** |
| | P-10 | **-8.7E+02** | **-8.7E+02** | **-8.7E+02** | -6.1E+02 | -6.9E+02 | 7.0E+02 | 2.6E+03 | -8.5E+02 |
| QUBO | P-11 | **-3.6E+02** | -3.5E+02 | -3.4E+02 | -3.2E+02 | -3.4E+02 | -3.2E+02 | -3.3E+02 | **-3.6E+02** |
| | P-12 | **-5.9E+03** | **-5.9E+03** | **-5.9E+03** | -5.6E+03 | **-5.9E+03** | -5.3E+03 | **-5.9E+03** | **-5.9E+03** |
| | P-13 | **-3.1E+00** | -3.0E+00 | -2.8E+00 | 0.0E+00 | 1.5E+01 | 2.8E+02 | -2.9E+00 | -3.0E+00 |
| | P-14 | **-3.1E+03** | -2.8E+03 | -3.0E+03 | -2.6E+03 | -3.0E+03 | -2.7E+03 | -3.0E+03 | -3.0E+03 |
| CONTROL | P-15 | **6.7E-03** | 7.4E-03 | 2.3E-02 | 8.4E-03 | 8.9E-03 | 3.1E-02 | 8.7E-02 | 7.3E-03 |
| | P-16 | **1.4E-02** | 2.6E-02 | 3.5E-02 | 1.7E-02 | 1.7E-02 | 5.3E-02 | 5.2E-02 | **1.4E-02** |
| | P-17 | **3.0E-02** | 5.7E-01 | 1.5E-01 | 4.8E-01 | 3.6E-02 | 7.7E-02 | 5.3E-02 | 3.7E-02 |
| CONTROL +CONSTR. | P-18 | 1.4E-02 | **1.1E-02** | 1.4E-02 | 3.4E-02 | 6.2E-02 | 2.8E-01 | 6.4E-02 | 2.1E-02 |
| | P-19 | **6.4E-02** | 5.7E-01 | 6.7E-02 | FAIL | FAIL | FAIL | FAIL | FAIL |
| | P-20 | **1.5E-01** | FAIL | 2.0E-01 | FAIL | FAIL | FAIL | FAIL | FAIL |

of these parameters was presented in Algorithm 1). For all methods, the limit on the number of requests to the objective function was fixed at the value $M = 10^4$. As seen from Table 1, PROTES, in contrast to alternative approaches, gives a consistently top result for almost all model problems (the best result for 19 of the 20 problems considered).

## 6.1 Multivariable analytic functions

First, we consider the optimization task for various tensors arising from the discretization of multivariable analytic functions. We select 10 popular benchmarks: Ackley (P-01), Alpine (P-02), Exponential (P-03), Griewank (P-04), Michalewicz (P-05), Piston[8] (P-06), Qing (P-07), Rastrigin (P-08), Schaffer (P-09) and Schwefel (P-10). These functions have a complex landscape and are often used in problems to evaluate the effectiveness of optimization algorithms [8, 16], including tensor-based optimizers [4, 39]. We consider the 7-dimensional case (since this is the dimension of the Piston function) and discretization on a uniform grid with 16 nodes.

As follows from Table 1 (benchmarks P-1, P-2, . . . , P-10), our method, like the other two tensor approaches (BS-1 and BS-2), gave the most accurate solution for all model problems. The most sophisticated approach from the nevergrad package (BS-7) was the next in accuracy (the method did not converge only in two cases out of ten).

## 6.2 Quadratic unconstrained binary optimization

QUBO is a widely known NP-hard problem [12] which unifies a wide variety of combinatorial optimization problems from finance and economics applications to machine learning and quantum computing. QUBO formulation in a very natural manner utilizes penalty functions, yielding exact model representations in contrast to the approximate representations produced by customary uses of penalty functions. The standard QUBO problem can be formulated as follows

$$\mathsf{f}(\boldsymbol{x}) = \boldsymbol{x}^T Q \boldsymbol{x} \to \min_{\boldsymbol{x}}, \quad s.t. \ \boldsymbol{x} \in \{0, 1\}^d,$$

where $\boldsymbol{x}$ is a vector of binary decision variables of the length $d$ and $Q \in \mathbb{R}^{d \times d}$ is a square matrix of constants. In all our experiments, we fixed the number of dimensions as $d = 50$.

---

[8]This function corresponds to the problem of modeling the time that takes a piston to complete one cycle within a cylinder; the description of its parameters can be found in [44, 4].

Table 2: Minimization results for benchmark P-12 for the 1000 dimensional case with different random initializations. Results for each value of $p$ are averaged over 5 runs of optimizers.

|  |  | PROTES | BS-1 | BS-2 | BS-3 | BS-4 | BS-5 | BS-6 | BS-7 |
|---|---|---|---|---|---|---|---|---|---|
|  | $p = 0.1$ | **-4.86E+05** | -4.40E+05 | FAIL | -4.10E+05 | -4.35E+05 | -3.94E+05 | -3.84E+05 | -4.71E+05 |
|  | $p = 0.2$ | **-9.70E+05** | -8.78E+05 | FAIL | -8.24E+05 | -8.71E+05 | -7.87E+05 | -7.64E+05 | -9.38E+05 |
| P-12 | $p = 0.3$ | **-1.46E+06** | -1.32E+06 | FAIL | -1.23E+06 | -1.30E+06 | -1.18E+06 | -1.14E+06 | -1.41E+06 |
|  | $p = 0.4$ | **-1.94E+06** | -1.75E+06 | FAIL | -1.64E+06 | -1.72E+06 | -1.57E+06 | -1.52E+06 | -1.88E+06 |
|  | $p = 0.5$ | **-2.43E+06** | -2.19E+06 | FAIL | -2.04E+06 | -2.16E+06 | -1.96E+06 | -1.90E+06 | -2.35E+06 |

We consider the following QUBO problems from the qubogen package:[9] Max-Cut Problem (P-11; which refers to finding a partition of an undirected graph into two sets such that the number of edges between the two sets is as large as possible), Minimum Vertex Cover Problem (P-12; which refers to finding a cover with a minimum number of vertices in the subset of the graph vertices such that each edge in the graph is incident) and Quadratic Knapsack Problem (P-13; which refers to finding a subset of maximum profit that satisfies the budget limitations from a set of potential projects with specified interactions between pairs of projects).

We also consider one more benchmark (P-14) from the work [11] (problem $k_3$; $d = 50$), where angle-modulated bat algorithm (AMBA) was proposed for high-dimensional QUBO problems with engineering application to antenna topology optimization. This is the ordinary binary knapsack problem with fixed weights $w_i \in [5, 20]$, profits $p_i \in [50, 100]$ $(i = 1, 2, \ldots, d)$, and the maximum capacity $C = 1000$. In experiments, we used the same values of the weights and profits as in [11].

For all four considered problems (P-11, P-12, P-13, P-14), the proposed method PROTES gives the best result, as seen from Table 1, and the baseline BS-7 again turned out to be the next in accuracy. We also note that several optimization methods were compared in [11] for the P-14 problem: BPSO (with the result $-2854$), BBA (with the result $-2976$), AMBA (with the result $-2956$), A-AMBA (with the result $-2961$), P-AMBA (with the result $-2989$), and the solution obtained using the PROTES method (the result $-3079$) turns out to be more accurate.

To study the stability of the proposed optimization method in the essentially multidimensional case, we also consider $d = 1000$ for the benchmark P-12. Additionally, we vary the value of the vertex connection probability in the generated random graph (i.e., the parameter in the corresponding function from the well-known networkx package), getting 5 different optimization problems with $p = 0.1$, $p = 0.2$, $p = 0.3$, $p = 0.4$, and $p = 0.5$ (note that in Table 1, we used the value $p = 0.5$). We performed the calculations with PROTES and baselines BS1 - BS7 with the same settings as above. In Table 2, we report the results averaged over 5 runs of each optimizer for each value of $p$. In all cases, PROTES demonstrates the best result. We note that BS2 failed for this problem because the selected dimension value is too large to construct the TT-approximation of the tensor.

In Figure 3, we plotted the convergence curves for $p = 0.1$, $p = 0.3$, $p = 0.5$, and all optimizers (except the BS2, since it failed), showing the average result and the spread of values over 5 restarts for P-12 ($d = 1000$). Our optimizer PROTES has a slight variance of the solution and consistently shows the best result when the number of iterations exceeds 2000.

## 6.3 Optimal control

Suppose we have a state variable $z \in \mathbb{R}$ controlled by a binary variable $x$ called control (i.e., it is just a switch with modes "off" = 0 and "on" = 1) over some discrete interval of time $[0, T]$. The state $z(t + 1)$ at time $t + 1$ depends on the control $x(t)$ at time $t$ and is obtained from the solution of the following differential equation $\dot{z}(\tau) = \mathsf{g}(z(\tau), x(t))$, $t \leq \tau < t + 1$, where the function $\mathsf{g}$ is called an equation function. The optimal control problem is to find such a sequence of controls $\boldsymbol{x}^* = [x^*(0), x^*(1), \ldots, x^*(T)]$ (optimal solution) over the given time interval $[0, T]$ that minimizes the given objective function $\mathsf{G}$.

---

[9]See https://github.com/tamuhey/qubogen.

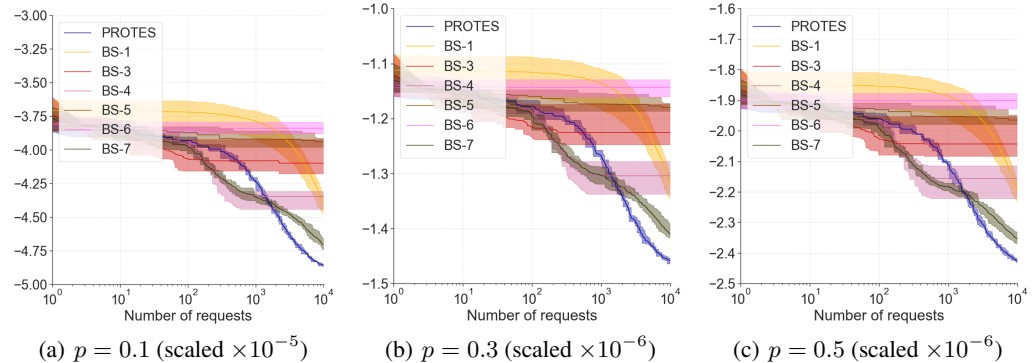

(a) $p = 0.1$ (scaled $\times 10^{-5}$)   (b) $p = 0.3$ (scaled $\times 10^{-6}$)   (c) $p = 0.5$ (scaled $\times 10^{-6}$)

Figure 3: Minimization results for benchmark P-12 in the 1000-dimensional case. The results for various values of the vertex connection probability $p$ of the generated random graph are presented. For each of the optimizers PROTES, BS1, and BS3 – BS7 (BS2 failed for this problem), we plot the value of the solution averaged over 5 runs with a solid line and fill in the area between the worst and best result with the same color.

Formulating the problem mathematically, we need to find such a solution

$$\mathsf{G}(\boldsymbol{z}, \boldsymbol{x}) \to \min_{\boldsymbol{z}, \boldsymbol{x}}, \quad \text{s.t.} \quad \begin{cases} z(0) = z_0, \\ \dot{z}(\tau) = \mathsf{g}(z(\tau), \boldsymbol{x}(t)), \ \ t \le \tau < t+1, \\ \boldsymbol{x}(t) \in \{0,1\}, \ \ t = 0, 1, \ldots, T, \end{cases}$$

where $\boldsymbol{z} = [z(0), z(1), \ldots, z(T)]$ is a state variable path. In numerical experiments, we consider the nonlinear equation function $\mathsf{g}(z, x) = z^3 - x$, and since it is nonlinear, finding an optimal solution raises a lot of difficulties. We take the objective function $\mathsf{G}$ in the form $\mathsf{G}(\boldsymbol{z}, \boldsymbol{x}) = \frac{1}{2} \sum_{t=0}^{T} (z(t) - z_{\mathrm{ref}})^2$. The initial and the reference state are fixed at values $z_0 = 0.8$, $z_{\mathrm{ref}} = 0.7$. For a fixed initial value $z_0$ and fixed equation function $\mathsf{g}$, the objective function $\mathsf{G}$ can be represented as a binary multidimensional tensor, whose elements are calculated using the following function: $\mathsf{f}(\boldsymbol{x}) = \mathsf{G}(\boldsymbol{z}(\boldsymbol{x}), \boldsymbol{x})$, hence we can apply discrete optimization methods to find $\boldsymbol{x}_{min}$, which approximates the optimal solution $\boldsymbol{x}^*$.

We considered several values for variable $T$, such as $25, 50$, and $100$ (benchmarks P-15, P-16, and P-17, respectively). As follows from the results presented in Table 1, PROTES gives the most accurate solution in all three cases. Note that a result comparable in accuracy to our method is obtained only in one case when using baselines (i.e., P-16, BS-7).

## 6.4 Optimal control with constraints

In practical applications, some conditions or constraints may be imposed on the solution of the optimal control problem. In this work, we consider the following control constraint $P$: *the control variable $\boldsymbol{x} \in \{0,1\}^N$ can take the value "1" no less than 3 times in a row during the whole time interval*. Formally, this can be written as follows:

$$P = \left\{ \boldsymbol{x} \ \middle| \ \begin{matrix} x[t] \ge x[t-1] - x[t-2] \\ x[t] \ge x[t-1] - x[t-3] \end{matrix}, \ \forall t : 1 \le t \le N+2; \text{we let } x[t] := 0 \text{ for } t < 1 \text{ and } t > N \right\}.$$

To account for this condition in the PROTES algorithm, we constructively build the initial distribution in the form of an indicator tensor as described in Section 4 in the constrained optimization subsection. The details of this construction are presented in the Appendix. The numerical results for $T = 25$ (P-18), $T = 50$ (P-19), and $T = 100$ (P-20) are reported in Table 1. In two cases out of three (P-19, P-20), our method showed the best result, and in one case (P-18) slightly yielding to the TTOpt method (BS-1), which, however, in two other cases, gave a significantly worse result.

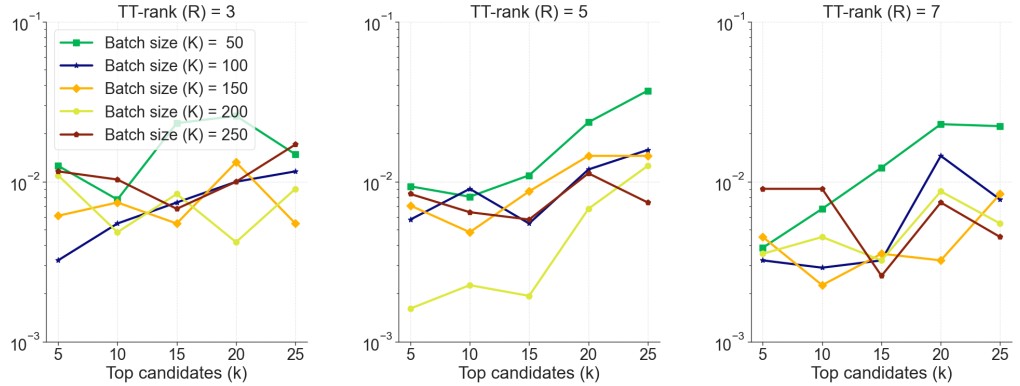

Figure 4: Relative error of the optimization result with PROTES method for the P-14 model problem with a known exact minimum for different values of the hyperparameters $K$, $k$ and $R$.

## 6.5 Robustness and performance of the PROTES

The results in Table 1 relate to the "intuitive" selection of the hyperparameters for the PROTES method (as was mentioned above, we have used the values: $K = 100$, $k = 10$, $k_{gd} = 1$, $\lambda = 0.05$, and $R = 5$). In Figure 4, we present an analysis of the dependence of the optimization result for the benchmark P-14 on the choice of hyperparameters $K$, $k$, and $R$, with fixed $k_{gd} = 1$ and $\lambda = 0.05$. We report the relative error of the result for all combinations $K = 50, 100, 150, 200, 250$; $k = 5, 10, 15, 20, 25$; $R = 3, 5, 7$. As we can see from the plots, the hyperparameters used in the main calculations are not optimal for this particular problem, that is, additional fine-tuning of the method for specific problems or classes of problems is possible. At the same time, according to the results in Figure 4, the method remains stable over a wide range of hyperparameter values. We also note that all computations were carried out on a regular laptop, while the operating time of the considered optimizers was commensurate; for example, for the benchmark P-17, the measured operating time (in seconds) is: PROTES (641), BS-1 (607), BS-2 (4245), BS-7 (780).

A more detailed analysis of the PROTES performance and the dependence of optimization results on the values of hyperparameters are presented in the Appendix. Also, in the Appendix, we consider another promising application of our optimizer for several popular reinforcement learning problems from Mujoco / OpenAI-GYM collection.

## 7 Conclusions

In this work, we presented an optimization algorithm PROTES based on sampling from the probability density defined in the tensor train format. We used the same set of hyperparameters for all considered numerical experiments, so our algorithm is rather universal To take into account the constraints, as in the problem of optimal control with constraints, we only considered them in the form of a specially selected initial approximation (a special form of an indicator tensor in the tensor train format); further on, the algorithm did not consider the constraints explicitly. This approach allows us to extend the algorithm's capabilities by using the properties of the tensor train representation. Numerical experiments show that we outperform many popular optimization methods.

The main direction in our future work is scaling the method to large dimensions. For $d \geq 1000$, we have encountered numerous technical difficulties, which can be alleviated by other tensor formats (such as hierarchical Tucker, which can be parallelized over $d$) and more efficient implementations of the optimization method (now we use standard automatic differentiation without special tensor optimization methods such as Riemannian optimization).

## Acknowledgements

The work is supported by the Ministry of Science and Higher Education of the Russian Federation (Grant 075-10-2021-068).

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

# Supplementary Material

## A1  Stability of the PROTES

To check the stability of the optimization result, a series of 10 calculations were performed for each method (PROTES, BS1 – BS7) with random initializations. We consider the binary knapsack problem from the work [11] (benchmark P-14, described in detail in the main text of the work) with the known exact minimum $-3103$. For all methods, the limit on the number of requests was fixed at the value $M = 10^5$. All other parameters were the same as in the computations from the main text, i.e., the PROTES parameters are $K = 100$, $k = 10$, $k_{gd} = 1$, $\lambda = 0.05$, $R = 5$. In Table 3 we present the average and best results over 10 runs for each optimization method. As follows from the reported results, only PROTES and Portfolio method (BS-7) managed to successfully find the exact optimum, while the average result for PROTES is significantly better than that of all the baselines.

Table 3:  Average and best result for 10 independent runs for the P-14 benchmark.

|       |      | PROTES | BS-1  | BS-2  | BS-3  | BS-4  | BS-5  | BS-6  | BS-7  |
|-------|------|--------|-------|-------|-------|-------|-------|-------|-------|
| P-14  | MEAN | **-3095** | -2992 | -3048 | -2650 | -2937 | -2701 | -3064 | -3075 |
|       | BEST | **-3103** | -3074 | -3084 | -2825 | -2996 | -2752 | -3094 | **-3103** |

## A2  Reinforcement learning with PROTES

As in the work [37], we consider maximization of the cumulative reward of the episode for several popular reinforcement learning problems from Mujoco / OpenAI-GYM collection: InvertedPendulum (P-21), Swimmer (P-22), Lunar Lander (P-23), and Half Cheetah (P-24). The policy is represented by a neural network with three layers and tanh activations. Parameters of the neural network are discretized on a grid with limits from $-1$ to $+1$ and 256 nodes (i.e., mode size $N$). The total number of neural network parameters for the problems under consideration turns out to be from 50 to 68 (i.e., dimension $d$). Thus, as in the work [37], we obtain a discrete on-policy learning problem (search for the values of the neural network parameters that lead to the maximum reward). To speed up the calculations, we took only two baselines (BS1, and BS7) for comparison, which showed the best results in the main experiments presented in the paper. We used the same optimizer settings as in the main calculations described in the text of the paper; however, in this case, we chose the budget value equal to $M = 10^5$, as in [37]. For PROTES, BS1, and BS7, we repeated the calculations three times and reported the average result for found maximum and computation time in Table 4. PROTES shows the best result in terms of accuracy in 3 cases out of 4, and once it is in second place, slightly inferior to the BS7.

Table 4:  Maximization results for InvertedPendulum (P-21), Swimmer (P-22), Lunar Lander (P-23) and Half Cheetah (P-24) benchmarks from Mujoco collection. We report the reward and calculation time (seconds), averaged over 3 runs, for the proposed method PROTES and baselines BS1 and BS7.

|            |      | REWARD |          |          | TIME    |         |         |
|------------|------|--------|----------|----------|---------|---------|---------|
|            |      | PROTES | BS-1     | BS-7     | PROTES  | BS-1    | BS-7    |
| RL AGENTS  | P-21 | **1.00E+03** | 7.28E+02 | 3.49E+02 | 8.1E+02 | 6.3E+02 | 1.3E+04 |
|            | P-22 | **3.61E+02** | 3.43E+02 | 3.55E+02 | 9.7E+03 | 7.7E+03 | 2.3E+04 |
|            | P-23 | **2.80E+02** | -1.87E+01 | 2.64E+02 | 3.5E+03 | 8.1E+03 | 1.5E+04 |
|            | P-24 | 2.26E+03 | 7.78E+02 | **2.66E+03** | 8.5E+03 | 5.5E+03 | 2.2E+04 |

## A3  Choice of hyperparameters for the PROTES

We conduct additional experiments with varying the values of the hyperparameters of the PROTES method for benchmarks P-01 (Figure 5 and Figure 6; we report the optimization result) and P-14 (Figure 4 in the main text and Figure 7 below; we report the relative error of the optimization result). In the first series of experiments (Figure 5 below and Figure 4 in the main text), we fixed the values $k_{gd} = 1$ and $\lambda = 0.05$ and tried all combinations of the remaining hyperparameters:

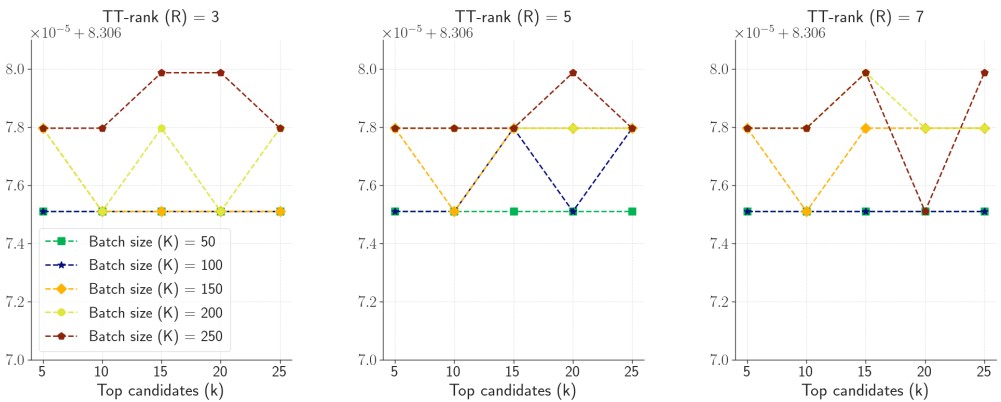

Figure 5: Optimization result (approximation of the minimum value) with PROTES method for the P-01 for different values of the hyperparameters: $K$, $k$ and $R$.

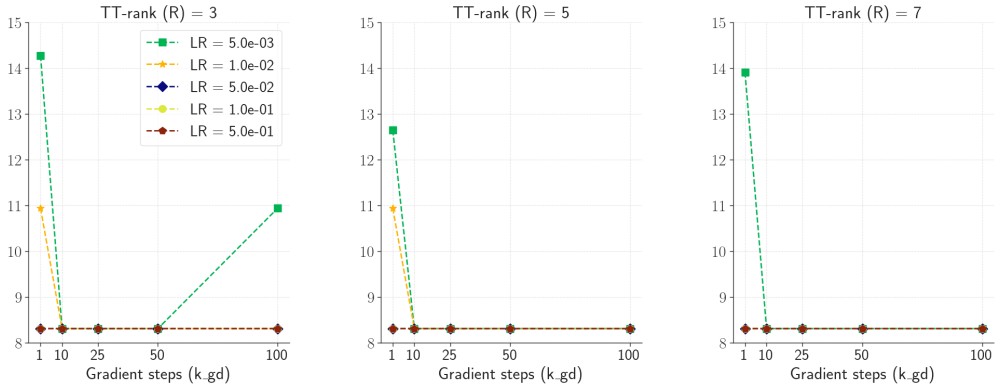

Figure 6: Optimization result (approximation of the minimum value) with PROTES method for the P-01 model problem for different values of the hyperparameters: learning rate (LR), $k_{gd}$, $R$.

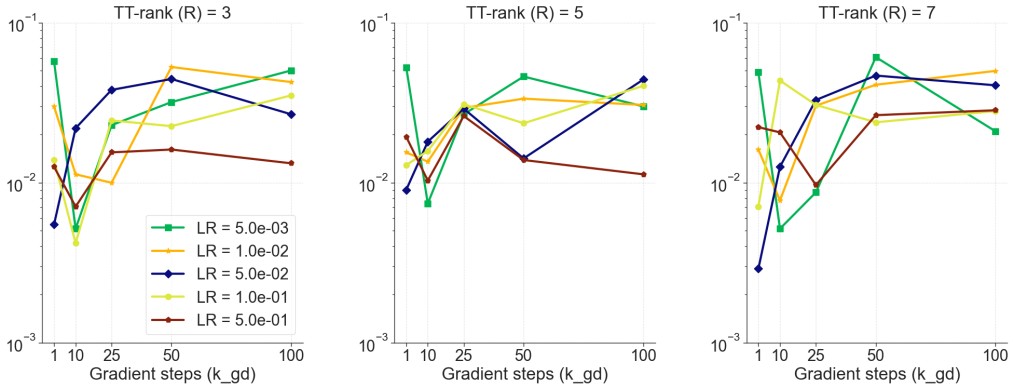

Figure 7: Relative error of the optimization result with PROTES method for the P-14 model problem with a known exact minimum for different values of the hyperparameters: learning rate (LR), $k_{gd}$, $R$.

$K = 50, 100, 150, 200, 250$; $k = 5, 10, 15, 20, 25$; $R = 3, 5, 7$. In the second series of experiments (Figure 6 and 7), we fixed the values $K = 100$ and $k = 10$, and tried all combinations: $k_{gd} = 1, 10, 25, 50, 100$, $\lambda = 0.005, 0.01, 0.05, 0.1, 0.5$; $R = 3, 5, 7$.

As follows from the presented results, for problem P-01, the dependence on the choice of hyper-parameters turns out to be extremely weak, except for outliers for the small values of the learning rate $\lambda = 0.005$ and $\lambda = 0.01$. For more complex problem P-14, the dependence of the result on the

Table 5: Computation time in seconds for all selected benchmarks (P-01 – P-20) and for all used optimization methods (PROTES, BS1 – BS7).

| | | PROTES | BS-1 | BS-2 | BS-3 | BS-4 | BS-5 | BS-6 | BS-7 |
|---|---|---|---|---|---|---|---|---|---|
| ANALYTIC FUNCTIONS | P-01 | 3.28 | 0.06 | 0.11 | 23.45 | 25.31 | 22.78 | 22.03 | 67.23 |
| | P-02 | 2.25 | 0.05 | 0.06 | 22.42 | 28.68 | 20.6 | 19.78 | 77.01 |
| | P-03 | 2.36 | 0.05 | 0.03 | 26.11 | 23.62 | 20.55 | 19.71 | 65.62 |
| | P-04 | 2.32 | 0.05 | 0.07 | 22.04 | 24.04 | 20.67 | 19.9 | 65.51 |
| | P-05 | 2.33 | 0.05 | 0.06 | 26.47 | 26.56 | 20.78 | 20.1 | 78.42 |
| | P-06 | 2.34 | 0.05 | 0.1 | 23.51 | 28.32 | 22.42 | 21.58 | 84.1 |
| | P-07 | 2.25 | 0.06 | 0.03 | 26.72 | 34.95 | 21.66 | 21.26 | 83.54 |
| | P-08 | 2.25 | 0.05 | 0.07 | 22.7 | 24.03 | 21.29 | 20.28 | 65.47 |
| | P-09 | 2.31 | 0.06 | 0.11 | 22.98 | 24.11 | 21.14 | 20.56 | 65.55 |
| | P-10 | 2.35 | 0.05 | 0.03 | 23.56 | 33.66 | 21.25 | 22.61 | 81.94 |
| QUBO | P-11 | 2.7 | 0.06 | 0.44 | 16.32 | 22.54 | 17.65 | 17.98 | 67.61 |
| | P-12 | 2.16 | 0.05 | 0.36 | 17.06 | 21.95 | 17.28 | 16.77 | 79.74 |
| | P-13 | 2.29 | 0.05 | 0.34 | 18.78 | 21.09 | 20.51 | 17.97 | 77.22 |
| | P-14 | 2.33 | 0.13 | 0.4 | 16.41 | 27.42 | 17.61 | 17.59 | 74.77 |
| CONTROL | P-15 | 513.6 | 1256.0 | 1839.0 | 550.0 | 545.0 | 620.3 | 530.4 | 707.5 |
| | P-16 | 542.4 | 969.4 | 3007.0 | 578.0 | 595.0 | 570.7 | 573.7 | 697.5 |
| | P-17 | 640.7 | 607.3 | 4245.0 | 673.6 | 687.6 | 661.0 | 687.1 | 779.5 |
| CONTROL +CONSTR. | P-18 | 328.1 | 92.9 | 588.1 | 319.1 | 516.2 | 202.9 | 533.9 | 474.5 |
| | P-19 | 8.74 | 69.49 | 912.0 | 17.13 | 17.45 | 16.16 | 16.84 | 47.43 |
| | P-20 | 9.23 | 0.53 | 931.8 | 21.16 | 21.42 | 20.28 | 20.84 | 62.26 |

choice of hyperparameters is more complex, but it can be seen that the method remains stable over a wide range of hyperparameter values.

## A4 Performance comparison of the optimization methods

All the results described in the main text and presented there in Table 1 were obtained on a regular laptop. In Table 5 we report the related computation time for each method (PROTES, BS1 – BS7) and each model problem (P01 – P20). As follows from the results, the PROTES works much faster than classical optimization methods (BS3 – BS7), as well as faster than tensor-based methods (BS1 and BS2) for several optimal control problems (P-15 – P-20). However, the TTOpt (BS-1) and Optima-TT (BS-2) methods are faster for simpler analytic (P-01 – P-10) and QUBO (P-11 – P-14) problems. This is due to the fact that within the framework of the dynamic TT-rank refinement procedure in these methods, the TT-rank, and hence the computation time, turn out to be significantly higher for the optimal control problems. We note that the very short running time of the TTOpt method for P-20 is because most of the requests of the method did not satisfy the imposed constraints, and in this case, the differential equation was not solved.

## A5 Derivative functions in the optimal control problem

We use the following derivative functions for the constructive building of the tensor described in [32]

$$f_0^k(x) = \begin{cases} 0, & x = 0 \text{ or } x = l \\ \texttt{None}, & \text{otherwise,} \end{cases}$$
$$f_1^k(x) = \min(l, \, x + 1),$$

for all TT-cores except the last one ($k = 1, 2, \ldots, d - 1$), and for the last TT-core

$$f_0^d(x) = \begin{cases} 1, & x = 0 \text{ or } x = l \\ 0, & \text{otherwise,} \end{cases}$$
$$f_1^d(x) = \begin{cases} 1, & x \geq l - 1 \\ 0, & \text{otherwise.} \end{cases}$$

A tensor in the TT-format built on such derivative functions is equal to $0$ if there are less than $l$ ones among its vector argument in a row, and is equal to $1$ in all other cases.

Let us briefly explain why such derivative functions give a tensor of the restriction condition (i.e., the constraint in the considered optimal control problem). Recall that the upper index in the derivative function notation corresponds to the index number of the tensor argument, and the lower index corresponds to the value of this argument index. In our scheme, the argument of the derivative functions has the meaning of the number of ones that already stand to the left of the given index.

First, let us focus on the functions for all indices except the rightmost one ($k = 1, 2, \ldots, d - 1$). If the current index is one, then we simply increase the value of the argument by $1$ and pass it on, to the input of the next derivative function. This is what the function $f_1^k$ does (we take the maximum, so if the number of ones has already reached $l$, we do not care if it is greater than or equal to $l$; note that the maximum operation reduces the TT-rank but does not affect the result). For the zero value of the current index, if the argument is zero (which means that the previous index is also zero), the function $f_0^k$ also returns zero as nothing have changed. If the argument of this function is $l$, it means that there are $l$ or more ones in a row to the left of the considered index, so the given condition is not violated, and the function simply returns zero, which means that there are no ones. If the argument is greater than zero but less than $l$, it means that the condition is violated because the previous sequence of ones of length less than $l$ is cut off at the current index. In this case, the function $f_0^k$ returns None, which means that the value of the tensor will be $0$ regardless of the subsequent indices.

The functions corresponding to the last ($k = d$) index behave similarly. Namely, if the last index is zero, the function $f_0^k$ returns $1$ if its argument is $0$ or $l$ since it does not violate the condition as just explained. Otherwise, it returns $0$. If the last index is $1$, the function $f_1^k$ returns $1$ only if its argument is equal to $(l - 1)$ or $l$ since this means that the current one is an element of a sequence of ones of length at least $l$. Otherwise, the function $f_1^k$ returns $0$.

