# OpenReview forum: "PROTES: Probabilistic Optimization with Tensor Sampling"
_NeurIPS.cc/2023/Conference — NeurIPS 2023 poster_

### Official Review · Reviewer_7wCj · 2023-07-05

**Soundness:** 2 fair
**Presentation:** 2 fair
**Contribution:** 2 fair
**Rating:** 5
**Confidence:** 4

**Summary:**

The authors present the algorithm PROTES for solving discrete optimization problems. The method uses probabilistic sampling to obtain a probability function of minimal (or maximal) values of a target function. This probability function takes the same shape as the target tensor. Alternatively, this could be described as a likelihood score for the optimal values of the discrete variables. To make the algorithm more efficient, the authors use the tensor train (TT)-representation for tensors. This is a low-rank approximation of a tensor, which makes the iterative computations faster. This TT-representation is used for the probability function.

The authors test their method on 20 different problem instances and compare their method to seven other methods. The results show that PROTES works well, and is stable for the instances it was tested on.

**Strengths:**

The paper has a clear structure and is well written.

**Weaknesses:**

The method was not well motivated and the update rule was unclear to the reader. This makes is it very hard to assess whether there is sufficient contribution to the research area.

The comment on the application of the method on constrained optimization is too short-sighted. In other words, such an important topic deserves more explanation. The current version does not clearly explain how general constrained optimization problems can be handled, particularly the feasibility concerns.

The instance sizes of the QUBO problems are very small compared to what is now possible with the constrained version of these problems.

Although the method was tested on 20 different problems, compared to seven benchmarks, only one instance per problem (size) was tested. For instance, for P-11 to P-13, it looks like only one instance was selected, which was not specified whether random problem parameters or something else was used as input. Therefore, the reader cannot tell if the proposed method works well in general for the tested problems or only on the tested instance.

**Questions:**

- What is the reason for using gradient ascent for updating the the probability distribution? Cannot alternatively a cheaper updating rule be used? Think for instance of increasing the likelihood of the selected candidates by some value in each iteration (similar to weighing).

- In line 62 of the paper, it is mentioned that the probability distribution is expected to present a Kronecker delta-function. It sounds more like an indicator function with a peak in the value of the minimum target function. Moreover, why is this function expected?

- Please address why only one instance was tested per problem (size), and why this is enough evidence of the efficacy of the method for the type of problem.

- In line 182, it is mentioned that related methods in the TT-format should be used for approximation. Is the approximation of the optimum meant here? It is a bit confusing since in the sentence before the (TT-)approximation of the original tensor is mentioned.

- What are the specifications (CPU etc) of the ‘regular laptop’?

**Limitations:**

Not applicabel.

---

> ### Author Rebuttal · Authors · 2023-08-08
>
> Thank you for the detailed analysis of our work and extensive comments and remarks! We tried to take them into account and make appropriate improvements. Below we present (within the specified limit on the length of the sent message) our responses to the main points of your review.
>
> 1. The method was not well motivated and the update rule was unclear to the reader.
>
>     - Indeed, as in the case of many other popular optimization methods, our approach is predominantly heuristic, so we confirm its effectiveness by comparing it with alternative known methods on a wide class of problems. Due to text size restrictions, we, unfortunately, could not describe in great detail all aspects. The scheme for updates we use (line 9 of Algorithm 1) is based on the automatic differentiation of our loss function from Formula 2, the rationale for which and connection with the REINFORCE trick is given in the section "The intuition behind the method". We note that due to the use of the tensor train (TT) decomposition, we can simultaneously efficiently (with the complexity linear in dimension) compute the likelihood for the automatic differentiation and sample from the probability tensor.
>
> 2. The comment on the application of the method on constrained optimization is too short-sighted.
>
>     - Due to the text size limitations, we could not describe this point in more detail, however, we will add additional comments to the Appendix. We note that even without explicitly taking into account the constraints (according to the scheme we described in the paper), our method is effective in comparison with baselines.
>
> 3. The instance sizes of the QUBO problems are very small...
>
>     - Thank you for this comment! We conducted a series of additional numerical experiments and described them in detail, as well as attached a PDF file with plots and tables in our "global response to all reviewers". In particular, we consider a 1000-dimensional QUBO problem (like P-12 in Table 1) in five different settings. As follows from the results in the PDF file, our optimizer PROTES demonstrates the best result. At the same time, we understand that there are specialized solvers for this class of problems that may be potentially more effective (at least in terms of speed), but we compared our general-purpose optimization method with similar non-specialized approaches.
>
> 4. ... only one instance per problem (size) was tested ...
>
>     - Thank you! We tried to take this remark into account by conducting a series of additional experiments (the results and a detailed discussion are presented in our "global response to all reviewers"). In particular, for the 1000-dimensional QUBO problem (like P-12 in Table 1) we consider 5 different options for initializing a random graph and also averaged the results for 5 independent runs of all optimizers.
>
> 5. What is the reason for using gradient ascent for updating the probability distribution?
>
>     - Since we are using a parametric representation (TT-decomposition) of the probability tensor, we can change the parameters, but not the value at one point or set of points (in this case, the change in the tensor turns out to be global, that is, a certain "learning" process takes place). We considered several variants close to that proposed by you (e.g., addition in the TT-format with subsequent rounding of the result in the TT-format, since the rank increases with addition), but we did not find significant improvement in the optimization results. Thus, we use automatic differentiation to update the TT-tensor parameters, since this is the most simple and common option, however, you are absolutely right that finding more efficient alternatives is a good direction for further research.
>
> 6. ... the probability distribution is expected to present a Kronecker delta-function ... why is this function expected?
>
>     - Thank you for this clarification about the indicator function. We expect this behavior, since when sampling, values closer to the optimum will more often be selected (after the first successful choice), under certain conditions for smoothness. We have tested this intuition with a number of examples and it was carried out, but it is obvious that certain counterexamples can be also found (strongly isolated optimum, multiple global optima, etc.). Also, due to budget constraints, this behavior is not fully realized. We have added relevant clarifications to the text of the paper.
>
> 7. In line 182, it is mentioned that related methods in the TT-format should be used for approximation...
>
>     - Thank you for this note! We have corrected the wording in the text of the paper. The Optima-TT essentially is a method for optimizing a surrogate model presented in the TT-format, that is, first, the TT-approximation (surrogate model) of the objective function is built using one of the existing approximation methods (TT-SVD, TT-ALS, TT-cross), and then the optimum of this model is searched with the Optima-TT. As we indicated in Footnote 5, when using this method as a baseline, we additionally applied the TT-cross method to build an approximation (surrogate model).
>
> 8. What are the specifications (CPU etc)?
>
>     - Numerical experiments presented in the paper were performed with the MacBook Pro (Intel Core i7, RAM 16 Gb). The wording "regular laptop" in the text meant that our method is not resource-intensive and does not require special computing equipment (however, it may certainly be needed in case of too complex objective functions for their efficient calculation).

---

> > ### Comment · Reviewer_7wCj · 2023-08-19
> > **Response to rebuttal**
> >
> > I thank the authors for their explanations to my questions. Due to the extra experiments and the well clarification on why gradient ascent is useful for this approach, I have raised my score.

---

> > > ### Author Response · Authors · 2023-08-19
> > >
> > > Thank you very much for your positive evaluation of our work! Your comments allowed us to improve the presentation of the proposal, and we thank you for that as well.

---

### Official Review · Reviewer_SmA2 · 2023-07-07

**Soundness:** 2 fair
**Presentation:** 3 good
**Contribution:** 2 fair
**Rating:** 6
**Confidence:** 2

**Summary:**

This paper presents a study on black-box optimization. The authors propose a new method based on sampling from the probability density defined in the tensor train (TT) format (extension effort). The experiments have shown that PROTES outperforms existing efforts.

**Strengths:**

(Overall) This paper is well organized. The technical soundness and practical performance are strong.

1. This paper has given a detailed motivation and an analysis of the computational complexity of the method. This makes the work appear comprehensive.

2. The research on gradient-free optimization plays a crucial role in various fields. Despite its significance, it often receives relatively low attention compared to gradient-based optimization methods. This effort tries to add distribution sampling for new tech, "TT-tensor."

**Weaknesses:**

1. The method section "Optimization method PROTES" is hard to follow and what's the relationship between this section and others?

2. The experiment (P1-P20) seems too simple to fully demonstrate the method's effectiveness.

3. This paper also uses the gradient ascent for the TT-tensor, so there lacks more discussion about the model design and gradient-free optimization.

**Questions:**

1. Does dimension d have an impact on the experiment? Why was it fixed at 50?

**Limitations:**

No negative societal impact.

---

> ### Author Rebuttal · Authors · 2023-08-08
>
> Thank you for your comments, which allowed us to improve the presentation of the proposed approach! Below we present our responses to your questions on the theoretical part of the work and comments on additional numerical experiments performed.
>
> 1. The method section "Optimization method PROTES" is hard to follow...
>
>     - We have done our best to make the presentation of the proposed method consistent: in section 2 we have given a simplified description of our approach with a graphical illustration; in section 3 we have briefly described the properties of the tensor train (TT) decomposition that underlies our method; and then in section 4 ("Optimization method PROTES"), we have given a formal description of the algorithm, an estimate of its computational complexity, and a possible connection to the REINFORCE trick. Unfortunately, due to page size limitations, we may not have been able to provide a very detailed description of all aspects. If the paper is accepted for the conference, we will of course try to improve the presentation of the method and will certainly take into account your recommendations.
>
> 2. The experiment (P1-P20) seems too simple...
>
>     - Thank you for this note! We conducted a series of additional numerical experiments and described them in detail, as well as attached a PDF file with plots and tables in our "global response to all reviewers". We consider four challenging problems of maximization of the cumulative reward of the episode for RL agents like InvertedPendulum, Swimmer, Lunar Lander, and Half Cheetah from Mujoco / OpenAI-GYM collection, and we consider the 1000-dimensional case for our benchmark P-12 (Minimum Vertex Cover problem).
>
> 3. This paper also uses the gradient ascent for the TT-tensor ... gradient-free optimization.
>
>     - Thank you to note this. We added a Footnote in the introduction on Line 34 ("...and updating its parameters to approximate the optimum...") explaining that our optimization method is gradient-free, i.e., it does not use the gradient of the objective function (black box) and also do not build an approximation of its gradient. At the same time, to update the internal parameters (elements of the TT-cores of the expectation tensor to the optimum presented in the TT-format), we use the gradient method. However, in our opinion, this fully preserves the correctness of classifying our method as gradient-free. We also especially note that in some cases, non-gradient approaches are possible for updating the expectation tensor (for example, we can use explicit formulas for the updates of the TT-cores).
>
> 4. Does dimension d have an impact on the experiment?
>
>     - We considered three classes of problems (the corresponding results were presented in Table 1): analytic functions (P-01 - P-10), QUBO problems (P-11 - P-14), and optimal control problems (P-15 - P-20; with and without constraints). For analytical functions, we have chosen dimension 7, since among them there is a Piston function (P-06) that is interesting from a practical point of view and has exactly this dimension. For optimal control problems, we considered the dimensions 25 (P-15 and P-18), 50 (P-16 and P-19), and 100 (P-17 and P-20); since our method significantly outperforms the baselines already at a dimension value of 100, we did not consider large values of the dimension. However, for QUBO problems, the chosen dimension value of 50 may indeed seem too low. We conducted a series of additional experiments for a dimension value of 1000 for QUBO problems. The results are presented in the attached PDF file. Due to the page limit, we report the result for only one of the problems, i.e., for P-12, but we considered various formulations of this problem corresponding to various options for initializing a random graph.

---

> > ### Comment · Reviewer_SmA2 · 2023-08-17
> >
> > Thank the authors for the clarifications. After reviewing the entire paper again, I tend to keep my current score.

---

> > > ### Author Response · Authors · 2023-08-19
> > >
> > > Thank you very much for your review and additional thorough analysis of our work! While "rebuttal", we did the best of ourself to fix the points you mentioned in the Weaknesses section. If you have any additional questions for us that may affect your assessment of our work, we will be happy to answer them within the remaining time for discussion. If not, we thank you once again for your positive evaluation of our proposal.

---

> > > > ### Comment · Reviewer_SmA2 · 2023-08-20
> > > >
> > > > Thanks for your effort during the entire rebuttal period. This work has been very solid, even without high novelty. After carefully reviewing the comments from the other reviewers, I would like to raise your score from a 5 to a 6.

---

> > > > > ### Author Response · Authors · 2023-08-20
> > > > >
> > > > > Thank you very much for raising the rating and positive evaluation of our proposal!

---

### Official Review · Reviewer_U74k · 2023-07-09

**Soundness:** 3 good
**Presentation:** 3 good
**Contribution:** 3 good
**Rating:** 6
**Confidence:** 3

**Summary:**

The paper proposes PROTES, a new method for multi-dimensinoal discrete black-box optimization based on adaptive probablistic sampling in the tensor train representation.

**Strengths:**

- The technical method is simple and intuitive, with a clear presentation.
- Diverse benchmark problems and comprehensive comparisons to relevant SOTA baselines.
- Good empirical performance under the tested experimental setup.

**Weaknesses:**

- Besides presenting and describing the results, lack of intuition/discussion on why the proposed approaches outperform baselines (both classical and tensor-based).
- It’s encouraged to have a wider spectrum of dimensionalities of the test problems besides 7 and 50 used.
- It's not the most efficient method for any given number of requests (evaluation budgets), as seen in Fig 3. The method gains more advantage as requests become closer to 10^4.

**Questions:**

N/A

**Limitations:**

The authors discussed the limitations of the paper which include the inability to handle large dimensions (>1000) and explicit constraint consideration.

---

> ### Author Rebuttal · Authors · 2023-08-08
>
> Thank you very much for your positive evaluation of our work and your comments! Below we present our responses; we hope that they will eliminate the shortcomings you have noted.
>
> 1. ... lack of intuition/discussion on why the proposed approaches outperform baselines ...
>
>     - A potential limitation of the considered tensor-based baselines (TTOpt and Optima-TT) is the requirement for the presence of a low-rank structure for the function being optimized. Since, within the framework of our method, a low-rank tensor approximation (tensor train decomposition) is constructed not for the objective function, but for the probabilistic expectation tensor of the optimum, our method potentially does not have this limitation (at the end of the optimization process, the probabilistic tensor is expected to be close to the "delta" function, which has exact TT-rank 1). At the same time, the classical baselines we have considered (PSO,  SPSA, etc.) are predominantly empirical methods, and our advantage can be explained by the successful sampling model used, which effectively combines exploration and exploitation. We also emphasize that the use of the low-rank representation (tensor train) is the cornerstone of our method since the number of parameters depends linearly on the dimension; for the TT-decomposition, there is an efficient sampling method with complexity linear in dimension; at the same time, tensor values can also be computed with complexity that is linear in dimension.
>
> 2. It’s encouraged to have a wider spectrum of dimensionalities of the test problems...
>
>     - Thank you for this comment! Indeed, in the original version of the manuscript, the dimensions of the problems did not exceed 100 (we considered 7-dimensional analytic functions; 50-dimensional QUBO problems; 25, 50, and 100-dimensional optimal control problems). We conducted a series of additional numerical experiments and described them in detail, as well as attached a PDF file with plots and tables in our "global response to all reviewers". This includes the more complex 1000-dimensional QUBO problem (like P-12 in Table 1), for which we performed computations for various values of the vertex connection probability in the generated random graph. In all calculations, our method outperforms the baselines.
>
> 3. The method gains more advantage as requests become closer to $10^4$.
>
>     - In the attached PDF file, we present convergence graphs for the new problem considered (1000-dimensional QUBO problem), from which the advantage of our approach is perhaps more noticeable. We also note that Table 1 in the Appendix, shows the results of multiple (10) runs of all optimizers for a specific problem with a known exact optimum value. As follows from the reported results, only our method and BS7 successfully find the exact optimum with 10 restarts, while the average result of our method is significantly better.
>
> 4. ... the inability to handle large dimensions (>1000) and explicit constraint consideration.
>
>     - As follows from the additional numerical experiments, our method remains efficient even when the dimension of the problem is 1000. However, in fact, for the case of higher dimensions, further development of our approach may be needed; in particular, the used TT-decomposition can be replaced by potentially more expressive and compact types of tensor networks, such as hierarchical Tucker decomposition, etc.

---

> > ### Comment · Reviewer_U74k · 2023-08-20
> >
> > Thank you for your response and additional experiments. Overall my concerns are mostly addressed, though it would be more informative to see full convergence curves rather than tables showing only the final performance, e.g. for Table 1 and 2 in the rebuttal pdf. The rebuttal didn't fully address this concern from my earlier review:
> > > It's not the most efficient method for any given number of requests (evaluation budgets), as seen in Fig 3. The method gains more advantage as requests become closer to 10^4.
> >
> > But given the speed efficiency and a good final performance at 10^4, along with the interesting idea, I remain positive and would like to keep my current score.

---

> > > ### Author Response · Authors · 2023-08-20
> > >
> > > Thank you for the positive assessment of our work and constructive comments, which allowed us to improve the quality of its presentation! We understand your remark on convergence, and we would like to make a small comment.
> > >
> > > According to the graphs for 1000-dimensional QUBO problems in the attached PDF file, our method becomes the most accurate starting from 2 thousand requests to the black box, and further there is a significant increase in the accuracy of the result (by the 10 K requests). In other words, for low computational budgets (providing low accuracy of the solution), our method (for the chosen fixed hyperparameters) does not exceed the BS-7, but, with an increase in the number of iterations, our method turns out to be much more accurate.
> > >
> > > However, you are absolutely right that the issues of convergence of our method (and the optimal choice of its hyperparameters) for significantly small computational budgets should be investigated as part of the further development of our approach. In particular, a heuristic for choosing the optimal batch size and top-k candidates depending on the available budget can be developed. Please note that in the current research, we used a single set of optimizer hyperparameters for all considered numerical problems.

---

### Official Review · Reviewer_e361 · 2023-07-24

**Soundness:** 3 good
**Presentation:** 3 good
**Contribution:** 3 good
**Rating:** 7
**Confidence:** 3

**Summary:**

This paper proposes a novel algorithm to estimate global extrema of discrete black-box target functions. The approach proposes learning a probability distribution, represented as a multidimensional tensor and efficiently stored using a TT-decomposition, that is trained to favor multi-indices that achieve high/low values in the target function. The approach, PROTES, draws inspiration from the REINFORCE algorithm, and can be trained using gradient-based optimization techniques. Extensive experiments are conducted to demonstrate the competitive performance and robustness of PROTES.

**Strengths:**

- The paper is quite self-contained and well written, concepts are introduced gradually and are easy to understand
- The connection to REINFORCE is informative and helpful
- Experiments are compelling, detailed ablation analyses are included demonstrating the robustness of PROTE


**Weaknesses:**

Clarity/Notation:
- "Gradient-free" is defined very late in the paper but is used frequently throughout the earlier sections. This is confusing since the optimization of the probability tensor is done with gradient based optimization. I would suggest providing a clear description of this in the introduction.
- It is stated in the footnote on page 1 that upper case letters are used to denote matrices, but they are used several times in the paper to refer to scalars i.e. $N_i$, $R_i$, etc.
- In figure 2, the tensor network diagram needs further explanation. It is not clear what is being conveyed from the figure and the current caption.

Comparison to Existing Approaches:
- While the experiments empirically demonstrate the efficacy of the proposed approach, the computation complexities of prior works is not stated anywhere in the paper. Is it the case that the performance benefits are attained by adding significant computational costs? This should be discussed in the comparison.
- Since the proposed approach is probabilistic, it would be useful to include error bars in the experimental section.


**Questions:**

- What are the computational complexities of the baselines?
- Does this method work especially well when there is a particular structure in the target function? Clearly this method, and any of the baselines would fail if $\mathcal{Y}$ is random.
- How is the probability tensor parametrized? Are the values of the tensor learnt directly? Could there be a benefit to alternative parametrizations (i.e. having a multilayer structure)?

**Limitations:**

Discussed in detail

---

> ### Author Rebuttal · Authors · 2023-08-08
>
> Thank you very much for the high appreciation of our work, very detailed comments and useful suggestions to improve its quality! Below we present our responses to the main points of your review.
>
> 1. "Gradient-free" is defined very late in the paper...
>
>     - You are absolutely right that this point should be clarified in the text. We added a Footnote in the introduction on Line 34 ("...and updating its parameters to approximate the optimum...") explaining that our optimization method is gradient-free, i.e., it does not use the gradient of the objective function (black box) and also does not build an approximation of its gradient. At the same time, to update the internal parameters (elements of the TT-cores of the expectation tensor presented in the TT-format), we use the gradient method. However, in our opinion, this fully preserves the correctness of classifying our method as gradient-free. We also especially note that in some cases, non-gradient approaches are possible for updating the expectation tensor (for example, we can use explicit formulas for the updates of the TT-cores), but we left this minor point outside the scope of this work.
>
> 2. ... upper case letters are used to denote matrices... used several times in the paper to refer to scalars...
>
>     - Thank you for noting this! We have added a comment in the Footnote 1 that lowercase and uppercase (where this does not lead to confusion) notation is used for scalars. Renaming uppercase letters like $N_i$ and $R_i$ is difficult since this is a common practice in tensor applications (e.g., $N_i$ is the maximum possible value, and $1 \leq n_i \leq N_i$ is a specific value). If you think that the current notation can make the text difficult to read, then we can, for example, use bold capital letters for matrices.
>
> 3. ... the tensor network diagram needs further explanation.
>
>     - Unfortunately, due to the limitation on the length of the text, we did not have the opportunity to make the necessary detailed explanations. We will add relevant explanations to the Appendix.
>
> 4. ... demonstrate the efficacy of the proposed approach ... What are the computational complexities of the baselines?
>
>     - Thank you for bringing attention to this important point. An estimate for the computational complexity of the proposed method was given in Formula 4 (Section "Computational complexity of the method"), and, under certain assumptions, it is close to the complexity estimates for the other two tensor methods chosen as baselines. Also, the Appendix contains the section "3 Performance comparison of optimization methods" with Table 2, which presents the measured running time of all methods for the considered benchmarks, and the corresponding discussion. It is important to note that the complexity of the optimization algorithm is largely determined by the complexity of the black box calculation, and if it is significant (and in most practical cases it is), then the determining factor is the number of performed requests to the black box, and not directly the complexity of the optimization algorithm inner operations. Since you have brought this to your attention, we will add a more detailed discussion in the appendix.
>
> 5. ...include error bars in the experimental section.
>
>     - Thank you for this comment! We conducted a series of additional numerical experiments and described them in detail, as well as attached a PDF file with plots and tables in our "global response to all reviewers". In particular, we presented plots for the mean and uncertainty in a PDF file for a more complex 1000-dimensional QUBO optimization problem that we additionally considered.
>
> 6. Does this method work especially well when there is a particular structure in the target function?
>
>     - This is a very important and interesting question, but, unfortunately, it is difficult to give an exact answer to it. As, for example, the authors of tensor optimizers TTOpt and Optima-TT, chosen by us as baselines, are expected their methods to work especially well if the function being optimized has a low-rank tensor structure. However, unlike these methods, we work with a low-rank approximation not of the objective function, but of the probability tensor, and it seems the constraint on the small rank is easier to fulfill in this case (i.e., we expect that the probability tensor will be close to the "delta function" as a result of optimization, and this function has the exact TT-rank equals 1).
>
> 7. How is the probability tensor parametrized?
>
>     - The probability tensor is represented in the form of a tensor train (TT) decomposition and its parameters are elements of three-dimensional TT-cores (as shown in Figure 2). The use of this parametric representation is the cornerstone of our method because, as discussed in the main text: 1) the number of parameters depends linearly on the dimension; 2) for the TT-decomposition, there is an efficient sampling method with complexity linear in dimension; 3) at the same time, tensor values can also be computed with complexity that is linear in dimension.
>
> 8. Could there be a benefit to alternative parametrizations?
>
>     - In view of our comment on the previous point, we consider the TT-decomposition to be an extremely good choice for the parametric representation of the probability tensor. However, you are right, there may be other options as well. In our opinion, it would be interesting to use orthogonalized TT-cores with a special update procedure (this will reduce the number of parameters) or the hierarchical Tucker decomposition, which, like the TT-decomposition, is a low-rank tensor approximation, but has a more complex (tree-like) structure, which can lead to greater expressiveness of the method.

---

> > ### Comment · Reviewer_e361 · 2023-08-17
> > **Response to Rebuttal**
> >
> > Thank you for your comprehensive rebuttal. All of my concerns have been addressed, so I have decided to raise my score.

---

> > > ### Author Response · Authors · 2023-08-19
> > >
> > > Thank you very much for the positive assessment of our work and the increase in the rating! We are very pleased that we were able to improve the quality of the presentation of the work based on your comments.

---

### Official Review · Reviewer_7NgC · 2023-08-01

**Soundness:** 2 fair
**Presentation:** 3 good
**Contribution:** 2 fair
**Rating:** 4
**Confidence:** 4

**Summary:**

The paper targets the challenging gradient-free discrete multidimensional optimization. It is formalized as finding a vector of d indices whose associated entry in a d-dimensional tensor is minimized, where the tensor contains all the possible values for the black-box objective function. There are an exponential number of entries in the tensor for brute-force search. Instead of enumerating all of them, the paper proposes to optimize a low-rank tensor whose each entry represents the probability of selecting the corresponding entry as the minimum in the objective value tensor. The algorithm starts from a random probability tensor and updates it using a similar trick as REINFORCE algorithm. In experiments on benchmark tasks, it performs comparably or better than the baselines evaluated.

**Strengths:**

1. It targets an important and challenging problem interesting to many practical problems.
2. The idea of using a low-rank tensor can reduce the cost of brute-force search.
3. Evaluation of different tasks and robustness to the hyperparameters.

**Weaknesses:**

1. It is a combinatorial search problem from an exponential number of candidates so some structural information of the problem has to be used. However, it is not clear why and when the low-rank probability tensor assumption holds, and if so, for which classes of problems.
2. The technical contribution is ad-hoc to some extent by combining tensor-train decomposition with REINFORCE trick.
3. I like the explanation of the intuition starting at line 122 but the approximation in Eq. (5) does not hold rigorously without using the true Monte-Carlo samples as in REINFORCE (the proposed alg instead uses the best k samples)
4. The complexity analysis assumes a fixed number of queries to the black-box function and does not study the number of iterations required to reach a sufficiently good local minimum.
5. The proposed method does not show advantages to BS-1 and BS-2 in Table 1 on analytic functions. Figure 3 does not show a clear advantage in the sample efficiency.

**Questions:**

1. Can you elaborate on when and why the combinatorial optimization problem's structure is consistent with the low-rank probability assumption? Some concrete examples can be helpful.
2. How to balance the exploration and exploitation in the black-box optimization problem? It seems like the proposed algorithm mainly focuses on exploitation (always selecting the top-k candidates).


**Limitations:**

The author did not discuss the potential limitations. An efficient solver for combinatorial black-box optimization can be used for attacks such as threats to cybersecurity. It would be nice to have some discussions on that.

---

> ### Author Rebuttal · Authors · 2023-08-08
>
> Thank you very much for your detailed comments and questions, which allowed us to refine the presentation of the proposed approach. Below we present our respective responses.
>
> 1. ... when and why the combinatorial optimization problem's structure is consistent with the low-rank probability assumption?
>
>     - We consider the case of black-box optimization, that is, knowledge of the internal structure of the target function is not assumed; and our optimizer shows good results on several classes of problems at once, unlike baselines. But we agree that the corresponding intuition, based on the knowledge of the internal structure, can be useful. As, for example, the authors of TTOpt and OptimaTT (BS1 and BS2 in our work), are expected their methods to work especially well if the target function has a low-rank tensor structure. However, unlike these methods, we work with a low-rank approximation not of the objective function, but of the probability tensor (P). We expect that the tensor P will be close to the "delta function" as a result of optimization, and this function has the exact TT-rank equals 1. At the beginning of the optimization process, the value of the selected rank determines the level of flexibility of the method, however, as shown in the Appendix, varying its value does not lead to a significant deterioration or improvement.
>
> 2. ... ad-hoc ... combining tensor-train decomposition with REINFORCE trick.
>
>     - Thank you for this comment, but we want to emphasize that our algorithm is not based on the REINFORCE trick. After the development and successful testing of the PROTES method, we saw some of its external similarities with the REINFORCE trick and discussed it in the section "The intuition behind the method". In particular, the loss function we use (please, see Formula 2) differs from formula 5 used in the REINFORCE trick.
>
> 3. ... does not hold rigorously without using the true Monte-Carlo samples as in REINFORCE ...
>
>     - In Formula 5, there is a functional F(f) containing parameters E and T. As we tried to explain in the text after this formula, with an appropriate choice of these parameters in the sum, just the top-k samples will make a noticeable contribution. On the other hand, indeed, with fixed values of the parameters E and T, the optimal number of selected top-k samples can change from iteration to iteration, but we did not see an increase in optimizer performance when using such a modification. We emphasize that our method is not directly based on the REINFORCE trick (the PROTES algorithm is significantly different), at the same time, of course, the study of the connection between the two approaches can give additional intuition.
>
> 4. ... not study the number of iterations required to reach a sufficiently good local minimum.
>
>     - We considered the case of a fixed budget as the most common one in our work (the value of "sufficient" accuracy is difficult to formalize and depends on the problem). We note that Table 1 in the Appendix, shows the results of multiple (10) runs of all optimizers for a specific task with a known exact optimum. As follows from the reported results, only our method and BS7 successfully find the exact optimum with 10 restarts, while the average result of our method is significantly better.
>
> 5. ... not show advantages to BS-1 and BS-2 in Table 1 on analytic functions...
>
>     - Yes, indeed, for this class of problems, all three tensor methods (PROTES, TTOpt, and Optima-TT) showed the best results, but for the other three classes of considered problems, the advantages of our approach, as follows from Table 1, are significant. The good result of TTOpt and Optima-TT for analytic functions may be because they have a small tensor rank, in contrast to more complex problems from other classes.
>
> 6. Figure 3 does not show a clear advantage in the sample efficiency.
>
>     - The demonstration on these graphs probably turned out to be not clear enough, since we chose not the best set of benchmarks for demonstration. We conducted a series of additional numerical experiments and described them in detail, as well as we attached a PDF file with plots and tables in our "global response to all reviewers". In the PDF file, we present new graphs of optimizer convergence (averaged over 5 restarts) for the considered more complex 1000-dimensional QUBO problem (like P-12 in Table 1). We hope that on these graphs, the advantage of our approach is more noticeable.
>
> 7. How to balance the exploration and exploitation?
>
>     - Thank you for this question! Within the logic of our algorithm, the balance is achieved through the optimal choice of the top-k parameter (that is, the number of good samples selected from the batch). If the value of the parameter is small, then mainly the points close to the current found local optimum are collected (exploitation). On the contrary, if the parameter value is large, then samples from a larger range can be selected, that is, the search area is expanded (exploration). We note that the Appendix contains an analysis of the dependence of the result of our optimizer on the choice of this parameter (as well as other parameters of the method).
>
> 8. ... potential limitations ... can be used for attacks such as threats to cybersecurity.
>
>     - We are categorically against the use of our developments for malicious purposes! Unfortunately, like the creators of other powerful tools in the field of optimization, machine learning, quantum physics, etc., we cannot significantly influence this process, since the principle of open source software is also important. To partially remove the risk you indicated, we have already limited in the license the possibility of commercial use of our software product without the explicit approval of the developers, with no restrictions on use for scientific purposes.

---

> > ### Comment · Reviewer_7NgC · 2023-08-18
> >
> > Thanks for the detailed response to my concerns! The rebuttal addressed some of my concerns but the major ones still remain after reading the rebuttal and the general response PDF. Here is some detailed feedback:
> >
> > - Re 1: This is still my major concern. Most combinatorial optimization problems have rich structures and solving them relies on such structures to shrink the search space. In addition, different problems have different structures. However, I do not see any theoretical analysis (even on some concrete examples of combinatorial optimization problems) in this paper showing a low-rank matrix of probabilities assigned to the candidate solutions is versatile and accurate enough to encode different structures. Low-rank approximation might be overly smooth for the approximation of many non-smooth combinatorial optimization problems.
> >
> > - Re 2: Eq. 2 is a log-likelihood and its gradient is computed using the same log-derivative trick of REINFORCE. Their algorithmic details are not exactly the same but I am not sure this computation of gradient can be a major novelty.
> >
> > - Re 3 & 7: It is good to know this is not a practical issue if the choices of E and T are correct. Empirical performance largely depends on the combinatorial problems included in the evaluation. But I am still not sure if it holds rigorously in theory and in general. The gradient computed based on the top-k samples can have a bias due to the lack of exploration on other candidates. This is also related to the exploration-exploitation trade-off mentioned in 7. Top-k selection is a greedy strategy so the exploration can be very weak unless you use a large k. But the cost of using such a large k is usually not affordable.
> >
> > - Re 4: This is a theoretical concern regarding the complexity analysis because assuming a fixed number of iterations cannot guarantee many favorable properties of the solution, e.g., local minimum, equilibrium, etc. This makes the conclusion of complexity very weak.
> >
> > - Re 5 & 6: Thanks for the explanation and experimental results in the PDF! They addressed some of my initial concerns. It seems like PROTES only outperforms baselines and shows better convergence speed in the very later stage. What happens in the earlier stages and how to explain this observation?
> >
> > - Re 8: This concern aims to point out that there does not exist a limitation section in the submission discussing the potential security risk of publishing such techniques proposed in the paper. It would be helpful to add such discussions to the paper.
> >
> > While I greatly appreciate the authors for their responses, my main concerns still remain on the main contribution/novelty and the theoretical correctness. In essence, this is a **low-rank approximation of the multi-linear extension in combinatorial optimization. But it is not convincing to me (both intuitively and theoretically) that this is guaranteed to be a good approximation in general.** Experiments did show the effectiveness of the method on some exemplar problems. But as I mentioned before, combinatorial optimization problems can be very different from each other. Hence, I will maintain my original score at this time.

---

> > > ### Author Response · Authors · 2023-08-19
> > >
> > > Thank you very much for your additional detailed comments. Below we present our responses to each of them [we split our response into two messages due to the text length limits].
> > >
> > > 1. [Re 1] About structures in the optimization problems.
> > >
> > >     - We fully agree that specialized solvers adapted for specific classes or types of problems (for example, there are a lot of specialized methods for QUBO problems) can be potentially more efficient than our optimizer on the corresponding class of problems. However, we have proposed a black box optimizer, and we compare its effectiveness with baselines operating in the same paradigm (PSO, SPSA, etc.). We note that baselines are also heuristic methods, but they have proven to be effective for many applied problems. Therefore, our goal was to consider several different classes of problems and show that our optimizer outperforms baselines on average over all the classes considered. As follows from the numerical experiments, for various analytic functions, QUBO problems, ODE-based optimal control and optimization of reinforcement learning agents, our method is stable and superior to alternative approaches. We will add relevant comments to the Conclusion section.
> > >
> > >     - We want to emphasize that we use the low-rank tensor train (TT) decomposition (with a fixed rank). It has a rich complex internal structure, and allows building accurate approximations for complex non-smooth functions, which is confirmed by many publications in various applied fields, including machine learning applications (we have provided some references in the text of the paper).
> > >
> > >     - At the same time, it is important to take into account that we do not approximate the objective function or loss function in the TT-format. We only use the approximation of the distribution, which can be sampled near the optimum of the objective function. Our hypothesis is that sufficient number of points closest to the optimum value of the target function lie on the low-rank manifold. Moreover, since the tensor changes with iterations, we do not approximate all the points closest to the current local optimum, but go along some "low-rank path" to the global optimum.
> > >
> > > 2. [Re 2] About the novelty and gradient computation.
> > >
> > >     - We want to note that we consider, in contrast to formula 5, an unweighted probability distribution, that is, we take top-k candidates and do not use the values of the target function. In addition to the empirical fact that this provides greater stability of the method, it can also be especially relevant in the presence of constraints on the objective function (values that do not satisfy the constraint can be easily discarded) or in the case of functions for which it is easy to select top-k candidates from a batch, but it is difficult to provide specific numerical values (for example, when choosing the best images from a set, etc.).
> > >
> > > 3. [Re 3 & 7] About the lack of exploration.
> > >
> > >     - We agree that selecting top-k candidates from a batch is consistent with a greedy strategy. However, when sampling from a probability distribution in the TT-format, we form a batch that will contain samples not only close to the current found local optimum, but also samples far from it ("random"), which relates to exploration. As follows from the numerical experiments, including the Table 1 in the Appendix, our method effectively finds a good approximation (and even exactly) of the global optimum.
> > >
> > > 4. [Re 4] About the fixed number of iterations.
> > >
> > >     - We want to note that in practical optimization applications, the budget is usually limited. Also, in many works, the evaluation of the effectiveness of methods is carried out with a fixed budget. In the main numerical experiments, we limited the budget for all considered optimizers to 10 K requests. Since we considered many different model tasks, it was difficult to use higher budget values for all tasks. However, as you noted ("About the convergence speed"), as the budget grows, the advantages of our method become even more significant.
> > >
> > >     - We also note that Table 1 in the Appendix, shows the results of multiple (10) runs of all optimizers for a specific problem with a known exact optimum value. As follows from the reported results, only our method and BS7 successfully find the exact optimum with 10 restarts, while the average result of our method is significantly better.

---

> > > > ### Author Response · Authors · 2023-08-19
> > > >
> > > > 5. [Re 5 & 6] About the convergence speed.
> > > >
> > > >     - According to the graphs in the file, our method becomes the most accurate starting from 1-2 thousand requests to the black box. We want to note that further there is a significant increase in the accuracy of the result (by the 10 K requests). In other words, for extremely low computational budgets (providing extremely low accuracy of the solution), our method (for the chosen fixed hyperparameters) does not exceed the BS-7, however, with an increase in the number of iterations, our method turns out to be much more accurate.
> > > >
> > > >     - We cannot conduct an additional numerical experiment within the framework of this discussion, however, we believe that by reducing the batch size for our method, its advantage will manifest itself at lower budget values (however, the final result on a large budget will be less accurate).
> > > >
> > > >     - It is also possible to give an intuitive explanation for this observation. The considered tensor-based methods (PROTES, TTOpt, Optima-TT) build some kind of internal model (TT-representation), and at the initial iterations until the model of sufficient quality is built, the sampling turns out to be of insufficient quality. Please pay attention to the behavior of the TTOpt method on these graphs - with small budgets, it has a significantly low accuracy, but then it starts to increase rapidly.
> > > >
> > > > 6. [Re 8] About the potential security risk.
> > > >
> > > >     - Thank you! We have included comments on this point in our original response. This is an important point, and we will add, as you recommend, appropriate explanations to the text of the paper.

---

### Author Rebuttal · Authors · 2023-08-08

Thank you very much for your comments, which allowed us to correct the shortcomings and refine the presentation of the proposed approach. We have improved the text of the manuscript, however, according to the rules of the conference, we are now unable to attach the updated version.

1. We practically did not meet major remarks from the Reviewers on the algorithmic part of the proposed new method PROTES for the multidimensional gradient-free optimization, but taking into account the comments, we will make three general replies on the theoretical part:

    - 1.1. In our approach, the probability tensor for the optimum expectation (P) is represented in the form of a low-rank (low-parametric) tensor train (TT) decomposition, and we iteratively optimize its parameters (i.e., the elements of three-dimensional TT-cores) to sample candidates for the optimum from this tensor. The use of this parametric representation is the cornerstone of our method since the number of parameters depends linearly on the dimension; there is an efficient sampling method in the TT-format with linear complexity; values of the TT-tensor can also be computed with linear complexity.

    - 1.2. Each step of updating the probability tensor turns out to be non-local: the probability increases not only for the selected best candidates ("top-k") but also the probabilities for all other multi-indices are redistributed within the framework of the parametric structure used. In this case, with a wide range of values of the "top-k" parameter, the balance between exploration and exploitation is maintained.

    - 1.3. We demonstrate the formal connection of our approach to the REINFORCE trick in the text, however, PROTES is not based on it. The algorithm and the loss function used are significantly different.

2. We tried to take into account the comments of the Reviewers on the experimental part of our work (i.e., insufficiently complex benchmarks; low values of dimensions for QUBO problems) by conducting a series of additional experiments. We report the results of our experiments in the attached PDF file (Table 1, Table 2, and Figure 1) and discuss them below:

    - 2.1. As in the work "TTOpt: A maximum volume..." (NeurIPS-2022), we consider four challenging problems of maximization of the cumulative reward of the episode for RL agents: InvertedPendulum (P-21), Swimmer (P-22), Lunar Lander (P-23), and Half Cheetah (P-24) from Mujoco / OpenAI-GYM collection. The policy is represented by a neural network with three layers and tanh activations. Parameters of the neural network are discretized on a grid with limits from -1 to +1 and 256 nodes (i.e., mode size N). The total number of neural network parameters for the problems under consideration turns out to be from 50 to 68 (i.e., dimension d). Thus, as in the work "TTOpt", we obtain a discrete on-policy learning problem (search for the values of the neural network parameters that lead to the maximum reward). To speed up the calculations, we took only two baselines (BS1, and BS7) for comparison, which showed the best results in the main experiments presented in the paper. We used the same optimizer settings as in the main calculations described in the text of the paper, however, in this case, we chose the budget value equal to M = 100 K, as in the work "TTOpt". For PROTES, BS1 and BS7, we repeated the calculations three times and reported the average result for found maximum and computation time in Table 1. As we can see, PROTES shows the best result in terms of accuracy in 3 cases out of 4, and once it is in second place, slightly inferior to the BS7. We propose to add Table 1 with a detailed text description to the new section-1 of the Appendix.

    - 2.2. We consider the 1000-dimensional case for our benchmark P-12 (Minimum Vertex Cover problem). Additionally, we vary the value of the vertex connection probability $p$ in the generated random graph (i.e., the parameter in the corresponding function from the well-known networkx package), getting 5 different optimization problems with p=0.1, p=0.2, p=0.3, p=0.4 and p=0.5 (please note that in the original text of the paper, we used the value p=0.5). We performed the calculations with PROTES and baselines BS1 - BS7 with the same settings as in the main computations described in the text of the paper. In Table 2 we report the results averaged over 5 runs of each of the optimizers for each value of p (we note that BS2 failed for this problem because the selected dimension value is too large to construct the approximation of the tensor). As we can see, in all cases our optimizer PROTES demonstrates the best result. We propose to add Table 2 with a detailed text description to the new section-2 of the Appendix.

    - 2.3. Using the numerical results from the previous paragraph (2.2), we plotted the convergence curves for p=0.1, p=0.3, p=0.5, and all optimizers (except the BS2, since it failed), showing the average result and the spread of values over 5 restarts. As we can see, our optimizer PROTES has a small result variance and consistently shows the best result when the number of iterations is more than 1000. We propose to add new Figure 1 to the new section-2 of the Appendix or replace Figure 3 in the main text with it.

We once again thank all the Reviewers for the positive assessment of our work, as well as for useful comments and advice that allowed us to improve its presentation! We will be happy to answer any additional questions you may have.

---

### Decision · Program_Chairs · 2023-09-21

**Decision:**

Accept (poster)

**Comment:**

This paper proposes a new method for black-box optimization over a sizable number of combinatorial (e.g., binary) parameters. Different to earlier work, the authors use an efficient tensor train (TT) representation, not for the unknown function, but to parameterize a distribution over it (i.e., a surrogate model).

This paper tackles an important and understudied problem in ML, where many BO methods use Gaussian process surrogate models, which do not work well for combinatorial spaces. The reviewers are also convinced about the empirical evaluations. Reviewers criticize that it is not clear which type of combinatorial problems are best approximated by the surrogate model here, but the authors convincingly argue that this is not an easy question to answer, but that the assumptions are certainly weaker than for prior work, where the function itself has to follow a TT structure (here, it is only the distribution over functions). Another criticism is that the problems in the experiments are fairly small in terms of number of parameters, but the authors post favourable results for another problem with 1000 variables. They also address criticism that the original submitted results are for single problem instances. I'd like to highlight the high quality and measuredness of the feedback, which seems to have satisfied the reviewers, leading to score increases (and myself as AC). If this paper gets accepted, the authors should carefully take the recommendations of the reviewers for more clarity into account. For example, there was some confusion about REINFORCE, whereas the authors seem to use automatic differentiation, so the relation to REINFORCE does not seem close.